# Softmax Transformers are Turing-Complete

**Hongjian Jiang**
RPTU Kaiserslautern-Landau
hongjian.jiang@rptu.de

**Michael Hahn**
Saarland University
mhahn@lst.uni-saarland.de

**Georg Zetzsche**
MPI-SWS
Kaiserslautern, Germany
georg@mpi-sws.org

**Anthony W. Lin**
RPTU Kaiserslautern-Landau and MPI-SWS
Kaiserslautern, Germany
awlin@mpi-sws.org

## ABSTRACT

Hard attention Chain-of-Thought (CoT) transformers are known to be Turing-complete. However, it is an open problem whether softmax attention Chain-of-Thought (CoT) transformers are Turing-complete. In this paper, we prove a stronger result that length-generalizable softmax CoT transformers are Turing-complete. More precisely, our Turing-completeness proof goes via the CoT extension of the Counting RASP (C-RASP), which correspond to softmax CoT transformers that admit length generalization. We prove Turing-completeness for CoT C-RASP with causal masking over a unary alphabet (more generally, for letter-bounded languages). While we show this is not Turing-complete for arbitrary languages, we prove that its extension with relative positional encoding is Turing-complete for arbitrary languages. We empirically validate our theory by training transformers for languages requiring complex (non-linear) arithmetic reasoning.

## 1 INTRODUCTION

Transformers (Vaswani et al., 2017) have enabled powerful Large Language Models (LLMs) with Chain-of-Thought (CoT) steps, which are capable of complex reasoning (cf. (Wei et al., 2022; OpenAI et al., 2024)). But *what task can (and cannot) be done by CoT transformers?* This fundamental question lies at the heart of the recent effort in understanding the ability of transformers through the lens of formal language theory (see the survey by Strobl et al. (2024)). In particular, the question whether CoT transformers is *Turing-complete* — that is, capable of solving any problems solvable by Turing machines — is especially pertinent; see the work (cf. (Pérez et al., 2021; Bhattamishra et al., 2020; Merrill & Sabharwal, 2024; Qiu et al., 2025; Li & Wang, 2025)).

**Are CoT transformers Turing-complete?** All existing proofs of Turing-completeness of CoT transformers (cf. (Pérez et al., 2021; Bhattamishra et al., 2020; Merrill & Sabharwal, 2024; Qiu et al., 2025; Li & Wang, 2025)) employ *hardmax attention*, which is a rather unrealistic assumption. In particular, its use comes at the cost of a lack of a trainability guarantee. It is still an open question to date whether CoT transformers that use softmax attention are Turing-complete, and whether one can guarantee some sort of trainability. A closer look at these proofs reveals a direct simulation of Turing machines using CoT transformers, where the position of the head of the Turing machine should be "deduced" by means of attention from the CoT tokens. This was so far achieved using averaging hard attention, which uses $-|\langle x, y \rangle|$ attention score (as in Pérez et al. (2021)) or layer norm (as in Merrill & Sabharwal (2024)). It is unclear how to achieve this using softmax; more generally, it is still an open question if softmax transformers can capture languages of averaging hard-attention transformers (see Yang & Chiang (2024); Yang et al. (2024)).

**Contributions.** The main contributions of this paper are (i) to prove for the first time that softmax CoT transformers are Turing-complete, and (ii) to provide a guarantee of length generalizability.

More precisely, we use the framework from Huang et al. (2025) of *length-generalizable* softmax transformers. Roughly speaking, a language $L$ is length generalizable if an idealized learning pro-

cedure (in the sense of Huang et al. (2025)) converges to $L$, if provided with all inputs of length $\leqslant i$ for some $i$. In particular, the authors showed that a simple declarative language called C-RASP (with causal masking) Yang & Chiang (2024) can be converted into their framework, thereby also admitting length generalization. To date, this is still one of the most predictive notions of trainability for transformers that have solid theoretical foundations, as well as extensive empirical evidence. Our results use the extensions of these models with CoT steps.

As we noted, a direct simulation of Turing machines using softmax transformers is rather tricky, as it would be challenging to extract the position of the head of the Turing machine by means of softmax attention. The main innovation in our proof technique is to exploit the *counting power* of softmax transformers (through C-RASP) to simulate *Minsky's counter machines*, instead of Turing machines. This would entail Turing-completeness of softmax transformers. The details of our results are below.

We first show that CoT C-RASPs with causal masking are Turing-complete over a unary alphabet $\Sigma = \{a\}$. More generally, we show that Turing-completeness holds for *letter-bounded languages*, i.e., $L \subseteq a_1^* \cdots a_n^*$, where $a_1, \ldots, a_n$ are distinct letters in the alphabet. Such languages are especially interesting because of their ability to model complex number-theoretic concepts (e.g., prime numbers, exponentiation, multiplication, etc.).

Interestingly, we show that CoT C-RASPs with causal masking are *not* Turing-complete over arbitrary languages. In fact, simple languages (e.g. palindromes) cannot be solved by CoT C-RASPs. To address this limitation, the next novelty in our proof is to extend CoT C-RASPs with *Relative Positional Encodings (RPEs)* (cf. Shaw et al. (2018); Liutkus et al. (2021); Dufter et al. (2022)), which assigns a positional information to any token *relative* to another token. We extend the framework of Huang et al. (2025) by adding RPEs, and show that length-generalizability still holds. Next, we provide a simple RPE that enables CoT C-RASP to work with arbitrary input words: they allow us to compute an unambiguous encoding of the input word into a number that can be accessed by the simulated counter machine. This results in full Turing-completeness in the presence of RPEs (surprisingly, in fact, a *single* RPE that works independently of the Turing machines).

We provide an experimental validation of our results for CoT C-RASP and CoT C-RASP[RPEs] by showing length generalization of transformers for complex number-theoretic concepts with *unary* representation (to be captured by CoT C-RASP) and with *binary* representation (to be captured by CoT C-RASP[RPEs]). For example, the concept of prime numbers will be represented as the language $L = \{a^p : p \text{ is prime}\}$ with unary representation, and as $L' = \{\text{bin}(p) : p \text{ is prime}\}$ with binary representation (where $\text{bin}(p)$ denotes the binary representation of $p$, e.g., 5 is written as 101).

**Organization.** We start with the CoT models in Section 2. We then prove Turing-completeness results for the unary and letter-bounded cases in Section 3. Turing-completeness for the general case is proven in Section 4. We report our experiments in Section 5. Finally, we conclude in Section 6.

## 2 MODELS FOR TRAINABLE COT TRANSFORMERS

### 2.1 TRANSFORMERS AND C-RASP

**Softmax Transformers.** We assume transformer decoders with softmax attention and causal masking (Softmax Attention Transformers, SMAT). Our formal definition of softmax transformers follows that of Huang et al. (2025). Attention weights are defined as

$$\bar{w} = \text{softmax}(\log n \cdot \{\mathbf{v}_j^T \mathbf{K}^T \mathbf{Q} \mathbf{v}_i\}_{j=1}^i) \tag{1}$$

where $\mathbf{v}_i$ denotes activations at position $i$, and $\mathbf{K}$, $\mathbf{Q}$ transform these to keys and queries, respectively. Here, scaling with $\log n$ is included, as it is needed to theoretically represent sparse functions across unboundedly input strings and circumvent theoretical limitations of soft attention (Chiang & Cholak, 2022; Edelman et al., 2022). For the feedforward networks, we assume one-layer networks, where each hidden unit has either ReLU or Heaviside activation. Here, as in Huang et al. (2025), Heaviside is needed to theoretically represent functions with sharp thresholds; at any finite input length, it can be arbitrarily closely approximated using ReLU MLPs. As is standard, we encode an input $x \in \Sigma^*$ by applying a token embedding function $\text{em} : \Sigma \to \mathbb{R}^k$ for some dimension $k$.

To define the computation of CoT via SMAT, we need the transformer to be able to output a token. We further define an output function $o : \mathbb{R}^d \to \Sigma$, parameterized by applying a linear function

$\mathbb{R}^d \to \mathbb{R}^{|\Sigma|}$ followed by an argmax selecting the symbol receiving the highest score. Overall, we view an SMAT as a length-preserving map $T : \Sigma^* \to \Sigma^*$, where $T(x)_i$ indicates the symbol predicted after reading the prefix $x_1 \ldots x_i$.

We refer to Appendix A for a formal definition and further discussion of design choices. We further refer to Appendix A.4 for a brief primer on the framework and results of Huang et al. (2025).

**C-RASPs.** C-RASP is equivalent to the fragment $\mathrm{K}_t[\#]$ Yang & Chiang (2024); Yang et al. (2024) of LTL[Count] Barceló et al. (2024) with only past operator:

$$\varphi \quad ::= \quad Q_a \, (a \in \Sigma) \mid \varphi \wedge \varphi \mid \neg\varphi \mid \varphi \vee \varphi \mid t \sim t \, (t \in \{<, =, >\})$$
$$t \quad ::= \quad c \, (c \in \mathbb{N}) \mid \overleftarrow{\#}[\varphi] \mid t + t$$

Let us define the semantics of C-RASP by structural induction on the C-RASP expressions. Suppose $w = w_1 \cdots w_n \in \Sigma^+$. [As a side remark, it is possible to also allow the empty string $\epsilon$ as input, and for this we can use the "start-of-string" symbol $\vdash$. We do not do this to avoid clutter.] For syntactic category $\varphi$, we will define $\llbracket\varphi\rrbracket_w$ as a bitstring $h_1 \cdots h_n \in \{0, 1\}^n$. On the other hand, for syntactic category $t$, we will define $\llbracket t\rrbracket_w$ as a sequence $m_1 \cdots m_n \in \mathbb{Z}^n$ of integers. For each sequence $\sigma$, we will write $\sigma(i)$ to denote the $i$th element in the sequence. We start with the two base cases:

- $\varphi = Q_a$. In this case, $h_i \in \{0, 1\}$ is 1 iff $w_i = a$.
- $t = c$. In this case, $m_i = c$ for each $i$.

We now proceed to the inductive cases:

- $\varphi = \psi_1 \wedge \psi_2$. Then, $h_i = \min\{\llbracket\psi_1\rrbracket_w(i), \llbracket\psi_2\rrbracket_w(i)\}$.
- $\varphi = \psi_1 \vee \psi_2$. Then, $h_i = \max\{\llbracket\psi_1\rrbracket_w(i), \llbracket\psi_2\rrbracket_w(i)\}$.
- $\varphi = \neg\psi$. Then, $h_i = 1 - \llbracket\psi\rrbracket_w(i)$.
- $\varphi = t \sim t'$. Then, $h_i = 1$ iff $\llbracket t\rrbracket_w(i) \sim \llbracket t'\rrbracket_w(i)$.
- $t = \overleftarrow{\#}[\varphi]$. Let $m_0 = 0$. Then, for each $i > 0$, $m_i = m_{i-1} + 1$ if $\llbracket\varphi\rrbracket_w(i) = 1$; else $m_i = m_{i-1}$.

**Relative Positional Encodings.** We also define an extension C-RASP[RPEs] (resp. SMAT[RPEs]) of C-RASP (resp. SMAT) with Relative Positional Encodings (RPEs), which are simply subsets $\mathfrak{R} \subseteq \mathbb{N} \times \mathbb{N}$. We start with C-RASP[RPEs]. In the sequel, the notation $\llbracket\mathfrak{R}\rrbracket$ refers to the function mapping each $(i, j) \in \mathbb{N} \times \mathbb{N}$ to $\{0, 1\}$ such that $\llbracket\mathfrak{R}\rrbracket(i, j) = 1$ iff $(i, j) \in \mathfrak{R}$. For the syntactic category $t$, we allow counting terms $\overleftarrow{\#}_{\mathfrak{R}}[\varphi]$ which is to be interpreted at position $j$ as the cardinality of $\{i \in [1, j] : (i, j) \in \mathfrak{R}, i \models \varphi\}$. Thus, we include $i$ depending on the positional encoding of each $i$ relative to $j$. [Alternatively, $\mathfrak{R}$ can be construed as allowing positions at certain distances from each $j$.] This generalizes the class C-RASP[periodic, local] defined by Huang et al. (2025), where $\mathfrak{R}$ is either periodic or local.

As for SMAT[RPEs], the definition is a simple modification of SMAT: the formula in (1) becomes

$$\bar{w} = \mathrm{softmax}(\log n \cdot \{\mathbf{v}_j^T \mathbf{K}^T \mathbf{Q} \mathbf{v}_i + \lambda \llbracket\mathfrak{R}\rrbracket(i, j)\}_{j=1}^i). \tag{2}$$

Here, we interpret $\lambda$ as a bias term and $\llbracket\mathfrak{R}\rrbracket(i, j)$ as 1 if $(i, j) \in \llbracket\mathfrak{R}\rrbracket$; otherwise, it is 0.

**Discussion of Relative Positional Encodings** Relative positional encodings, which modify attention scores with positional information, are a popular approach for providing positional information to transformers. Our formalization of RPEs is a simple formal abstraction of *additive relative positional encodings*, which add a position-dependent term to the attention logits (Shaw et al., 2018; Dai et al., 2019; Xue et al., 2021; Press et al., 2022; He et al., 2021). Schemes in the literature differ in whether they are parameter-free (e.g., Press et al. (2022)) or involve learnable parameters. We consider the especially simple case where $R$ is determined a-priori, parameter-free, and independent of the task at hand. We provide more discussion in Appendix A.3.

## 2.2 Extensions with Chain-of-Thought

Suppose $\Gamma$ is the (finite) set of possible CoT tokens. CoT tokens in some $\Gamma_F \subseteq \Gamma$ are reserved to indicate that the computation is to terminate and that the input string is to be "accepted". Let $\Gamma_{\neg F} = \Gamma \backslash \Gamma_F$. We define a CoT to be a map $F : \Sigma^* \to \Gamma^* \cup \Gamma^\omega$, where $\Gamma$ is a finite set of CoT tokens, where all non-final symbols are in $\Gamma_{\neg F} \subseteq \Gamma$. Here, note that we include both finite (terminating) CoTs in $\Gamma^*$ and infinite (non-terminating) CoTs in $\Gamma^\omega$. Consideration of non-terminating CoTs is needed for Turing completeness. The language $L(F)$ recognized by $F$ is the set of all $w \in \Sigma^*$ where $F(w)$ is finite and ends in an element of $\Gamma_F \subseteq \Gamma$.

**CoT C-RASPs.** We extend C-RASP (resp. C-RASP[RPEs]) with CoTs as follows. A CoT C-RASP expression (over $\Gamma$) is a non-empty sequence $S = d_1, \dots, d_l$ of definitions $d_i$ of the form:

$$O_{a_i} \leftarrow \varphi_{a_i},$$

where $a_i \in \Gamma_{\neg F}$ and $\varphi_{a_i}$ a normal C-RASP (resp. C-RASP[RPEs]) expression. The intuition of $S$ is a *switch* condition, which will tell the program which token to output. $S$ *outputs a token* on an input string $w \in (\Sigma \cup S)^+$ if $[\![\varphi_{a_i}]\!]_w(|w|) = 1$ for some $i$. The *output* of $S$ on a string $w \in (\Sigma \cup \Gamma_{\neg F})^+$ is defined to be $a_i$, where $i$ is the smallest index such that $[\![\varphi_{a_i}]\!]_w(|w|) = 1$ and that $[\![\varphi_{a_j}]\!]_w(|w|) = 0$ for each $j < i$. In this case, we write $S(w) = a_i$. Note that a CoT transformer might terminate without outputting a token if $[\![\varphi_{a_j}]\!]_w(|w|) = 0$ for each $j$; in this case, the input string $w$ will be immediately rejected. Here, we write $S(w) = \bot$ (i.e. undefined).

A CoT C-RASP $S$ *generates the string* $U = U_1 \cdots U_m \in \Gamma^*$ *on the input* $w \in \Sigma^*$ if $S(wU_1 \cdots U_{k-1}) = U_k$ for each $k = 1, \dots, m$. Intuitively, this means that $S$ autoregressively outputs the symbols in $U$. The *language $L(T)$ accepted by a CoT C-RASP $S$* is defined to be the set of all $w \in \Sigma^*$ such that there exists a finite string $U \in \Gamma^*$ ending in an element of $\Gamma_F$ such that $T$ generates $U$ on $w$, and non-last symbols in $U$ are in $\Gamma_{\neg F}$.

We remark that, in many cases, the order of the sequence $S$ is not so important, especially if we can ensure that at most $O_{a_i}$ is going to be satisfied. We will use this in the sequel.

**CoT SMATs.** Recall that we view an SMAT $T$ as a length-preserving map $T : \Sigma^* \to \Sigma^*$, where $T(x)_i$ indicates the symbol predicted after reading the prefix $x_1 \dots x_i$. An SMAT $T : (\Sigma \cup \Gamma)^* \to (\Sigma \cup \Gamma)^*$ *generates the string* $U = U_1 \cdots U_m \in \Gamma^*$ *on the input* $w$ if $T$ autoregressively predicts the string $U$ – that is, if $T(wU_1 \cdots U_{k-1}) = U_k$ for each $k = 1, \dots, m$. The *language $L(T)$ accepted by a CoT SMAT $T$* is defined to be the set of all $w \in \Sigma^*$ such that there exists a finite string $U \in \Gamma^*$ ending in $\Gamma_F$ such that $T$ generates $U$ on $w$, and non-last symbols in $U$ are in $\Gamma_{\neg F}$

**Proposition 2.1.** *If a language is accepted by a CoT C-RASP (resp. C-RASP[RPEs]), then it is also accepted by a CoT SMAT (resp. SMAT[RPEs]).*

*Proof Sketch for Proposition 2.1; see Appendix A.2 for full details.* The starting point is Theorem 9 in Huang et al. (2025), which shows that C-RASP can be simulated by *limit transformers*, which in turn are closely related to SMAT[RPEs]. This earlier result concerned language acceptance by a single binary label computed at the final token; we extend it to CoT generation, obtaining a SMAT that at each position outputs a one-hot vector indicating which CoT token to output. $\square$

## 2.3 Learnability with CoT

We now show that CoT C-RASP is learnable in the framework of Huang et al. (2025). Intuitively, this framework considers transformers being trained on data from some bounded length and then deployed on data of larger lengths. We now make this formal. As before, we view SMATs as defining length-preserving maps $T : \Sigma^* \to \Sigma^*$. The *hypothesis class* $\Theta$ is the set of SMATs $T$ where each parameter vector and matrix of $T$ is represented at $p$ bits of precision, for some $p$ depending on $T$.

**Definition 2.2.** *A language $L$ is* length-generalizably learnable with CoT *if there is a CoT $F$ with $L(F) = L$ such that the following holds: For each $i = 1, 2, 3, \dots$, use the idealized learning procedure from Definition 6 in Huang et al. (2025) to choose a sequence of SMATs $T_i \in \Theta$ (i =*

$1, 2, 3, \ldots$ ) *such that each $T_i$ generates $F(w)_{1 \ldots i - |w|}$ on all inputs $w$, $|w| \leqslant i$.*[1] *Then, there is some $N_0$ depending on $L$ such that for all $i > N_0$, $T_i$ will exactly recognize the language $L$ with CoT.*

For the purpose of understanding the rest of the paper, the details of the idealized learning algorithm from Definition 6 of Huang et al. (2025) is not of utmost importance, though suffice it to say that it attempts to minimize a regularizer that results in favoring simpler and smaller transformers. Interested readers can find more details in Appendix A.4.

Next, we analogously define the same notions in the presence of RPEs. Given a set $\mathfrak{R} \subseteq \mathbb{N} \times \mathbb{N}$, define the hypothesis class $\Theta[\mathfrak{R}]$ as the set of SMAT[RPEs] $T$ with the RPE $\mathfrak{R}$, where each parameter vector and matrix of $T$ is represented at $p$ bits of precision, for some $p$ depending on $T$, and where each $\lambda$ in (2) is fixed to 1. We then define *length-generalizably learnable with CoT with RPE $\mathfrak{R}$* by replacing $\Theta$ with $\Theta_{\mathfrak{R}}$ in Definition 2.2.

Here, the intuition is that we can learn a single SMAT that works for all input lengths, even when training only on data from some bounded length, as long as the training length is sufficiently large. We note that the definition of the learning setup is substantially simpler than in Huang et al. (2025) since our transformers use no absolute positional encodings. Whereas Huang et al. (2025) used separate hypothesis classes $\Theta_n$ at each context window size $n$, our learning setup requires a single hypothesis class $\Theta$ that works for all input lengths. We then obtain the following guarantee:

**Proposition 2.3.** *Consider a language expressible in C-RASP[RPEs] CoT, using RPE $\mathfrak{R}$. Then it is length-generalizably learnable with RPE $\mathfrak{R}$.*

*Proof Sketch for Proposition 2.3; see Appendix A.2 for full proof.* The proof is a straightforward adaptation of results of Huang et al. (2025). Theorems 7 and 9 in that paper show length-generalizable learnability for languages expressible in C-RASP without CoT. Building on Proposition 2.1, we extend this to CoT C-RASP. $\square$

# 3 UNARY CASE

In this section, we prove Turing-completeness of of CoT SMAT for unary alphabet, i.e., $\Sigma = \{a\}$. More precisely, CoT SMAT recognizes all recursively enumerable languages over unary alphabet. In fact, we prove stronger Turing-completeness results for letter-bounded languages and permutation-invariant languages. In turn, these results will be proven by establishing CoT C-RASPs for such languages and invoking Proposition 2.1. To help with readability, the reader may see Example 1, where we construct a CoT C-RASP for the PARITY language, which is incidentally known (cf. Huang et al. (2025)) not to be expressible by C-RASP without CoT.

**Theorem 3.1.** *Each recursively enumerable language over a unary alphabet $\Sigma = \{a\}$ can be recognized by SMAT in the CoT setting.*

The theorem follows from the following proposition and Proposition 2.1.

**Proposition 3.2.** *Each recursively enumerable language over a unary alphabet $\Sigma = \{a\}$ can be recognized by C-RASP in the CoT setting.*

In turn, this follows directly from the following proposition; recall that a language $L \subseteq \Sigma^+$ is *letter-bounded* if it is a subset of $a_1^* a_2^* \cdots a_n^*$ for some distinct letters $a_1, \ldots, a_n \in \Sigma$.

**Proposition 3.3.** *Each recursively enumerable letter-bounded language over any alphabet $\Sigma$ can be recognized by C-RASP in the CoT setting.*

We will deduce Proposition 3.3 from the following proposition, which will be most convenient for our construction. Given an alphabet $\Sigma$ with $\Sigma = \{a_1, \ldots, a_n\}$, the corresponding *Parikh map* is the map $\Psi \colon \Sigma^* \to \mathbb{N}^n$, where $w \in \Sigma^*$ is mapped to $(|w|_{a_1}, \ldots, |w|_{a_n})$, where $|w|_{a_i}$ is the number of occurrences of $a_i$ in $w$. In other words, $\Psi(w)$ is the vector that contains all letter counts in $w$. Notice that for $u, v \in \Sigma^*$, we have $\Psi(u) = \Psi(v)$ if and only if $v$ can be obtained from $u$ by re-arranging the letters, or by *permuting* $u$. We say that a language $L \subseteq \Sigma^*$ is *permutation-invariant* if for any $u, v \in \Sigma^*$ with $\Psi(u) = \Psi(v)$, we have $u \in L$ if and only if $v \in L$. In other words, membership in $L$ does not depend on the order in which letters appear in a word.

---

[1]Such a sequence always exists, as there is just a finite number of inputs at each length $i$.

**Proposition 3.4.** *Each recursively enumerable permutation-invariant language over any alphabet $\Sigma$ can be recognized by C-RASP in the CoT setting.*

We prove Proposition 3.4 by simulating counter machines. To define these, we define $\Phi_k$ to the set of expressions $\varphi$ of the following form: a conjunction of counter tests of the form $x_i \sim 0$, where $x_i$ indicates the $i$th counter and $\sim \in \{>, =\}$. A *$k$-counter machine ($k$-CM)* is a tuple $(P, \Delta, q_0, F)$, where $P$ is a set of states, $\Delta \subseteq P \times \Phi_k \times P \times \mathbb{Z}^k$ is a finite set of *transitions*, $q_0 \in P$ is the *initial state*, and $F \subseteq P$ is the set of final states. We also assume that the machine is *deterministic*, i.e., for any transitions $(p, \varphi, q, \boldsymbol{u})$ and $(p, \varphi', q', \boldsymbol{u}')$ starting in the same state $p$, but with $(q, \boldsymbol{u}) \neq (q', \boldsymbol{u}')$, the expressions $\varphi$ and $\varphi'$ cannot hold at the same time (i.e. $\varphi \wedge \varphi'$ is unsatisfiable). For a transition $\tau = (p, \varphi, q, \boldsymbol{u})$, we will use the notation $\mathrm{src}(\tau) := p$, $\mathrm{tgt}(\tau) := q$, $\varphi_\tau := \varphi$, and $\boldsymbol{u}_\tau := \boldsymbol{u}$.

A *configuration* of such a $k$-CM is a tuple $(q, \boldsymbol{x}) \in P \times \mathbb{Z}^k$, where $q \in P$ and $\boldsymbol{x} \in \mathbb{Z}^k$. For configurations $(p, \boldsymbol{x}), (q, \boldsymbol{y}) \in P \times \mathbb{Z}^k$, we write $(p, \boldsymbol{x}) \to (q, \boldsymbol{y})$ if there is a transition $\tau \in \Delta$ with $\mathrm{src}(\tau) = p$, $\mathrm{tgt}(\tau) = q$, $\varphi_\tau(\boldsymbol{x})$ is true, and $\boldsymbol{y} = \boldsymbol{x} + \boldsymbol{u}_\tau$. By $\xrightarrow{*}$, we denote the reflexive transitive closure of the relation $\to$ on the configurations. A configuration $(q, \boldsymbol{x})$ is *initial* if $q = q_0$. We say that an initial configuration $(q_0, \boldsymbol{x})$ is *accepted* if $(q_0, \boldsymbol{x}) \xrightarrow{*} (p, \boldsymbol{y})$ for some $\boldsymbol{y} \in \mathbb{Z}^k$ and $p \in F$. In other words, if there exists a run of the $k$-CM that eventually arrives in a final state.

We will employ the following variant of the fact that counter machines are Turing-complete. Note that if one uses CM as language acceptors, with input-reading transitions, then just two counters are sufficient for Turing-completeness. In our construction, it will be most convenient to provide the input of the CM at its counters. In this setting, it is known that three additional counters (aside from the input counters) are sufficient for Turing-completeness:

**Lemma 3.5.** *For every recursively enumerable set $S \subseteq \mathbb{N}^n$, there is a $(n+3)$-CM so that for every $\boldsymbol{x} \in \mathbb{N}^n$, the configuration $(q_0, \boldsymbol{x}, 0, 0, 0)$ is accepted if and only if $\boldsymbol{x} \in S$.*

This is a direct consequence of CM, as language acceptors are able to recognize all recursively enumerable languages (this is implicit in (Minsky, 1961, Theorem Ia), and explicit in (Fischer et al., 1968, Theorem 3.1)) and that $k$-CM accept the same languages as 3-CM (Greibach, 1976, Theorem 2.4). Moreover, if $S \subseteq \mathbb{N}^n$ is recursively enumerable, then the language $L := \{a_1^{x_1} \cdots a_n^{x_n} \mid (x_1, \ldots, x_n) \in S\}$ is a recursively enumerable language, and

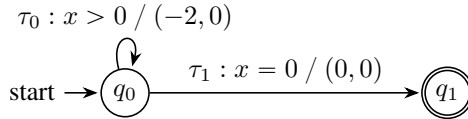

$\tau_0 : x > 0 \,/\, (-2, 0)$

$\tau_1 : x = 0 \,/\, (0, 0)$

start $\to q_0 \quad\quad q_1$

Figure 1: 2-CM with transition labels $\tau_i$.

so there exists a three-counter machine $M$ that recognizes $L$. This three-counter machine can easily be turned into a $(n+3)$-CM as we need it: whenever $M$ reads a letter $a_i$, our CM will decrement the $i$-th counter; and when $M$ uses counter $j \in \{1, 2, 3\}$, then our CM will use counter $n + j$.

**Corollary 3.6.** *For every recursively enumerable permutation-invariant language $L \subseteq \Sigma^+$, there is a $(n+3)$-CM so that for every $w \in \Sigma^+$, we have $w \in L$ if and only if $(q_0, \Psi(w), 0, 0, 0)$ is accepted.*

*Proof.* Follows from Lemma 3.5: For a recursively enumerable $L \subseteq \Sigma^+$, the Parikh image $\Psi(L)$ is recursively enumerable; since $L$ is permutation-invariant, we have $w \in L$ iff $\Psi(w) \in \Psi(L)$. $\quad\square$

*Proof of Proposition 3.4.* Let $\Sigma = \{a_1, \ldots, a_n\}$ and take a permutation-invariant recursively enumerable language $L \subseteq \Sigma^*$. From Corollary 3.6, we get a $(n+3)$-CM such that from the configuration $C_0 := (q_0, x_1, \ldots, x_n, 0, 0, 0)$, the CM will reach $F$ if and only if $a_1^{x_1} \cdots a_n^{x_n} \in L$.

We define the set $\Gamma$ of CoT tokens to be $\Sigma$ unioned with the transition relation $\Delta$. Note that the C-RASP is going to be evaluated at the last position on input $wv$ where $v \in \Gamma^*$. The construction of the C-RASP CoT transformer considers the following cases.

**Initial step.** At the beginning, the last symbol in the input to the C-RASP is in $\Sigma$. This indicates that the CM is in the initial state $q_0$. We add the following rules to our CoT C-RASP expression $S$

$$O_\tau \leftarrow \varphi(\overleftarrow{\#}[Q_{a_1}], \ldots, \overleftarrow{\#}[Q_{a_n}]) \wedge Q_a,$$

for each $a \in \Sigma$ and each transition $\tau = (q_0, \varphi, q', \boldsymbol{u}) \in \Delta$. The order in which the rules are added is not important since the counter machine is deterministic.

**Non-initial step.** After an initial step, the last symbol in the input is always a transition of the CM, which indicates which state the CM is in. We add the following rules to our CoT C-RASP expression $S$ (in no particular order):

$$O_{\tau'} \leftarrow \varphi_{\tau'}(t_1, \ldots, t_{n+3}) \wedge Q_\tau,$$

for any $\tau, \tau' \in \Delta$ with $\mathrm{tgt}(\tau) = \mathrm{src}(\tau')$. Here, $t_1, \ldots, t_{n+3}$ are the count-valued C-RASP terms

$$t_i = \overleftarrow{\#}[Q_{a_i}] + \sum_{\rho \in \Delta} \boldsymbol{u}_\rho(i) \cdot \overleftarrow{\#}[Q_\rho] \qquad \text{for } i = 1, \ldots, n \tag{3}$$

$$t_i = \sum_{\rho \in \Delta} \boldsymbol{u}_\rho(i) \cdot \overleftarrow{\#}[Q_\rho] \qquad \text{for } i = n+1, n+2, n+3. \tag{4}$$

Intuitively, each $[\![t_i]\!]_w$ will tell us the value of the $i$th counter. For $i = 1, \ldots, n$, we have the additional summand $\overleftarrow{\#}[Q_{a_i}]$ because this is the initial value of the $i$th counter, according to Lemma 3.5.

**Output symbols.** The desired output symbols for acceptance are any $\tau \in \Delta$ for which $\mathrm{tgt}(\tau) \in F$.

**Correctness.** The C-RASP directly simulates the CM, so correctness is immediate. □

Finally, Proposition 3.3 follows easily from Proposition 3.4: We can modify our C-RASP to check (e.g. in each step) that the (initial) input word belongs to $a_1^* \cdots a_n^*$. See Appendix B for the proof.

**Example 1.** *In this example, we illustrate the construction of CoT C-RASP for parity (i.e. $\{w \in \{a, b\}^+ : |w|_a \equiv_2 0\}$), which is a permutation invariant language. Note that this was proven not to be expressible in C-RASP without CoT Huang et al. (2025). We start with the the two 2-counter machine as depicted in Figure 1. To make the illustration simpler, we have opted to use only 2 counters (which are sufficient for this language), instead of 5 counters. The counter machine starts at $(q_0, x, y)$, where $x$ records the number of $a$'s and $y$ the number of $b$'s. It reduces $x$ by 2 until $x$ becomes zero, at which point it accepts by moving to $q_1$.*

*We now specify the C-RASP rules for the counter machine. We use $c$ as an arbitrary letter in $\{a, b\}$. We start with initial step, corresponding to the first transition taken by the counter machine:*

$$O_{\tau_0} \leftarrow \overleftarrow{\#}[Q_a] > 0 \wedge Q_c \qquad\qquad O_{\tau_1} \leftarrow \overleftarrow{\#}[Q_a] = 0 \wedge Q_c \tag{5}$$

*Note that, for our language, acceptance is only possible when the input is nonempty, i.e., the last symbol at the initial step is some $c \in \{a, b\}$. The C-RASP for the non-initial steps are as follows:*

$$O_{\tau_0} \leftarrow \overleftarrow{\#}[Q_a] - 2 \cdot \overleftarrow{\#}[Q_{\tau_0}] > 0 \wedge Q_{\tau_0} \qquad O_{\tau_1} \leftarrow \overleftarrow{\#}[Q_a] - 2 \cdot \overleftarrow{\#}[Q_{\tau_0}] = 0 \wedge Q_{\tau_0} \tag{6}$$

## 4 GENERAL CASE

Given that Propositions 3.3 and 3.4 show that for letter-bounded or permutation-invariant languages, CoT C-RASP are Turing-complete, this raises the question of whether they are even Turing-complete for a general language $L \subseteq \Sigma^*$. The following shows that they are not:

**Proposition 4.1.** *C-RASP in the CoT setting is not Turing-complete over $\Sigma = \{a, b\}$.*

This follows from the following lemma (e.g. take PALINDROME).

**Lemma 4.2.** *If a language $L$ is recognized by CoT C-RASP, then for each $n$ the restriction $L_n \subseteq L$ to all inputs of length $\leqslant n$ is recognized by an automaton of size polynomial in $n$.*

This is an immediate corollary of the logarithmic communication complexity of Limit Transformers and hence C-RASP (Theorem 12 in Huang et al. (2025)), which also holds even equipped with *learned Absolute Positional Encodings (APEs)*. See Appendix C for details. However, we will show that there is *one* simple relative positional encoding that makes CoT C-RASP fully Turing-complete:

**Theorem 4.3.** *Every recursively enumerable language over an arbitrary alphabet $\Sigma$ can be recognized by C-RASP[RPEs] in the CoT setting and, thus, can be recognized by CoT SMAT[RPEs].*

Membership in CoT SMAT[RPEs] follows from Proposition 2.1.

The CoT C-RASP[RPEs] constructed in Theorem 4.3 is based on the following idea. Given an input $w \in \Sigma^*$ with say $|\Sigma| = n$, our CoT C-RASP[RPEs] first computes an encoding of $w \in \Sigma^*$ as a vector in $\mathbb{N}^n$. After this, it uses a construction similar to above to simulate a CM on this encoding.

To avoid confusion between multiplication of $0$ and $1$ on the one hand and concatenation of words, we will use different symbols for the numbers $0, 1 \in \mathbb{N}$ and the letters $\mathtt{0}$ and $\mathtt{1}$. Then for $a = 0$, $b = 1$, $c = \mathtt{0}$, and $d = \mathtt{1}$, we can distinguish between $ab = 0$ and $cd = \mathtt{01}$. To convert between these objects, we use the notation $\overline{0} := \mathtt{0}$, $\overline{\mathtt{0}} = 0$, $\overline{1} = \mathtt{1}$, and $\overline{\mathtt{1}} = 1$.

**Encoding words over two letters**  We first describe how to encode two-letter words. Formally, we have a partial function $\beta \colon \mathbb{N} \rightarrow \{\mathtt{0}, \mathtt{1}\}^*$, where $\rightarrow$ means that $\beta$ is partial, i.e. not every number represents a word. However, if a number does represent a word, then that word is unique. A number $x \in \mathbb{N}$ will represent a word if and only if $x \neq 0$. Hence, suppose $x \neq 0$. Then we can write $x = \sum_{i=0}^{m} b_i 2^i$, where $b_m, \ldots, b_0 \in \{0, 1\}$, and $b_m = 1$. Let $j = \max(\{i \in [0, m] \mid b_i = 0\} \cup \{0\})$ be the left-most position of a zero when writing the most significant digit first (and $j = 0$ if zero does not appear). Then we set
$$\beta(x) := \overline{b_{j-1}}\, \overline{b_{j-2}}\, \cdots\, \overline{b_0}.$$
In other words, $\beta(x)$ is the word consisting of all digits of $x$'s binary representation, when reading from most significant digit first, and starting after the left-most zero digit (and $\beta(x) = \varepsilon$ if there is no zero digit). For example, we have
$$\beta(2^5 + 2^3 + 2^1) = \mathtt{1010}, \qquad \beta(2^6) = \mathtt{00000}, \qquad \beta(2^4 + 2^3 + 2^1) = \mathtt{10}.$$

**Encoding words over arbitrary alphabets**  Now suppose $\Sigma$ is an arbitrary alphabet with $\Sigma = \{a_1, \ldots, a_n\}$. Then we encode words in $\Sigma^*$ by vectors in $\mathbb{N}^n$. Similar to above, we define a partial function $\sigma \colon \mathbb{N}^n \rightarrow \Sigma^*$. Let us first describe the domain of $\sigma$. We say that an $n$-tuple $(w_1, \ldots, w_n)$ of words $w_1, \ldots, w_n \in \{\mathtt{0}, \mathtt{1}\}^*$ is *consistent* if (i) the words $w_1, \ldots, w_n$ have the same length, say $m \in \mathbb{N}$ and (ii) for every position $i \in [1, m]$, there is exactly one $j \in [1, n]$ such that $w_j$ has the letter $\mathtt{1}$ at position $i$. Intuitively, the consistent $n$-tuples correspond exactly to the words in $\Sigma^*$: A word $w \in \Sigma^*$ of length $m$ corresponds to the $n$-tuple $(w_1, \ldots, w_n)$ where each $w_i$ has length $n$, and the $\mathtt{1}$'s in $w_i$ are exactly at those positions that carry $a_i$ in $w$. This leads to an intermediate partial function $\mu \colon (\{\mathtt{0}, \mathtt{1}\}^*)^n \rightarrow \Sigma^*$, where $\mu(w_1, \ldots, w_n)$ is defined if and only if $(w_1, \ldots, w_n)$ is consistent, and in that case, $\mu(w_1, \ldots, w_n) \in \Sigma^*$ is the word corresponding to $w_1, \ldots, w_n$.

With this, we are ready to define $\sigma$. The domain of $\sigma$ consists of those $\boldsymbol{x} = (x_1, \ldots, x_n) \in \mathbb{N}^n$ where (i) all entries are non-zero and (ii) the tuple $(\beta(x_1), \ldots, \beta(x_n))$ is consistent. Moreover, for $\boldsymbol{x} = (x_1, \ldots, x_n) \in \operatorname{dom} \sigma$, we set
$$\sigma(\boldsymbol{x}) := \mu(\beta(x_1), \ldots, \beta(x_n)).$$
For example, for $n = 2$, we have
$$\sigma(2^4 + 2^0, 2^4 + 2^2 + 2^1) = \mu(\beta(2^4 + 2^0), \beta(2^4 + 2^2 + 2^1)) = \mu(\mathtt{001}, \mathtt{110}) = a_2 a_2 a_1.$$

An important property of $\sigma$ is that if we change $\boldsymbol{x} = (x_1, \ldots, x_n)$ by introducing further $\mathtt{1}$'s on the left of the binary representation of some $x_i$, then $\sigma(\boldsymbol{x})$ remains the same. For example, we also have
$$\sigma(2^5 + 2^4 + 2^0, 2^4 + 2^2 + 2^1) = \mu(\beta(2^5 + 2^4 + 2^0), \beta(2^4 + 2^2 + 2^1)) = \mu(\mathtt{001}, \mathtt{110}) = a_2 a_2 a_1.$$

although we modified the left-most entry by introducing the term $2^5$. Thus, for every $w \in \Sigma^*$ and every $k \in \mathbb{N}$, there is an $\boldsymbol{x} \in \mathbb{N}^n$ such that (i) all entries in $\boldsymbol{x}$ are $\geq k$ and (ii) $\sigma(\boldsymbol{x}) = w$.

**The relative positional encoding**  A key ingredient in our proof is the relative positional encoding (recall that we have shown that without RPE, Theorem 4.3 does not hold). Perhaps surprisingly, the RPE we use in the proof does *not* depend on the language we are accepting: *the RPE is the same relation for every Turing machine we want to simulate*. Its definition is based on the partial function $\beta \colon \mathbb{N} \rightarrow \{\mathtt{0}, \mathtt{1}\}^*$ above. We define the relation $\mathfrak{R} \subseteq \mathbb{N} \times \mathbb{N}$ as
$$(i, j) \in \mathfrak{R} \iff i \leq j, i \in [1, |\beta(j)|], \text{ and the word } \beta(j) \in \{\mathtt{0}, \mathtt{1}\}^* \text{ has } \mathtt{1} \text{ at position } i$$
for every $(i, j) \in \mathbb{N} \times \mathbb{N}$. For example, if $j = 2^6 + 2^5 + 2^3 + 2^1 + 2^0$, then we have $\beta(j) = \mathtt{1011}$ and hence $(1, j), (3, j), (4, j) \in \mathfrak{R}$, but $(2, j) \notin \mathfrak{R}$.

**Overview** Our C-RASP with CoT will work in *two phases*. During the *first phase*, it prolongs the input so that subsequently, a $\sigma$-encoding of the original input word can be computed using Count-Valued Operations. For this, it relies on the RPE $\mathfrak{R}$. In the *second phase*, our C-RASP simulates a counter machine, similar to the permutation-invariant case.

**Phase I: Constructing encoding of the input word** In order to compute the $\sigma$-encoding $\boldsymbol{x} \in \mathbb{N}^n$ of the input word $w \in \Sigma^*$, our CoT C-RASP proceeds as follows. It compute the entries $\boldsymbol{x}(1), \ldots, \boldsymbol{x}(n)$ of $\boldsymbol{x}$ in this order. Suppose $(w_1, \ldots, w_n)$ is the consistent tuple representing $w$, i.e. $\mu(w_1, \ldots, w_n) = w$. To compute $\boldsymbol{x}(1)$, our CoT C-RASP appends a dummy letter $\square_1$ until the current word length $\ell$ satisfies $\beta(\ell) = w_1$. Note that this is possible since there are infinitely many $\ell$ with $\beta(\ell) = w_1$. Once this holds, we place a special letter $\boxplus_1$. Then, the CoT C-RASP appends a dummy letter $\square_2$ until the current word length satisfies $\beta(\ell) = w_2$, and then places $\boxplus_2$, etc.

Initially, the last letter will be some $a_i \in \Sigma$. Then, our CoT C-RASP simply outputs $\square_1$: We have

$$O_{\square_1} \leftarrow Q_{a_i} \tag{7}$$

for each $a_i \in \Sigma$. When we have a letter $\square_i$ at the end, our CoT C-RASP checks whether the current length $\ell$ already satisfies $\beta(\ell) = w_i$:

$$O_{\boxplus_i} \leftarrow Q_{\square_i} \wedge \overleftarrow{\#}_{\mathfrak{R}}[Q_{a_i}] = \overleftarrow{\#}[Q_{a_i}] \wedge \overleftarrow{\#}_{\mathfrak{R}}[\top] = \overleftarrow{\#}[Q_{a_i}] \tag{8}$$

$$O_{\square_i} \leftarrow Q_{\square_i} \wedge (\overleftarrow{\#}_{\mathfrak{R}}[Q_{a_i}] \neq \overleftarrow{\#}[Q_{a_i}] \vee \overleftarrow{\#}_{\mathfrak{R}}[\top] \neq \overleftarrow{\#}[Q_{a_i}]) \tag{9}$$

for each $i = 1, \ldots, n$. If we evaluate rule 8 on a word of length $\ell$, we check that (i) the last letter is $\square_i$, (ii) the number of positions $j$ with $(j, \ell) \in \mathfrak{R}$ that carry $a_i$ equals the total number of positions that carry $a_i$, and (iii) the number of positions $j$ with $(j, \ell) \in \mathfrak{R}$ equals the number of positions that carry $a_i$. Thus, conditions (ii) and (iii) say that the positions $j$ with $(j, \ell) \in \mathfrak{R}$ are precisely those that carry an $a_i$. In other words, $\beta(\ell) = w_i$. If these conditions are met, then the output letter is $\boxplus_i$.

Moreover, if we evaluate rule 9, we check that $\beta(\ell)$ does not equal $w_i$ yet. In this case, the output letter is again $\square_i$, and the whole check will be repeated with the next word length.

If the last letter is $\boxplus_i$ with $i \leqslant n - 1$, then we start computing $\boldsymbol{x}(i + 1)$: We output $\square_{i+1}$ in 10:

$$O_{\square_{i+1}} \leftarrow Q_{\boxplus_i} \qquad \text{for each } i = 1, \ldots, n-1 \tag{10}$$

$$O_\tau \leftarrow Q_{\boxplus_n} \qquad \text{for each transition } \tau \in \Delta \text{ with } \mathrm{src}(\tau) = q_0 \tag{11}$$

If the last letter is $\boxplus_n$, we initiate the CM run by outputting some initial transition $\tau$. This is rule 11.

After the above process, we have placed $\boxplus_1, \ldots, \boxplus_n$. Thus, the current input word is then of the form $w' = w\square_1^{f_1} \boxplus_1 \square_2^{f_2} \boxplus_2 \cdots \square_n^{f_n} \boxplus_n$, where for the tuple $\boldsymbol{x} = (x_1, \ldots, x_n)$ with $x_i = |w| + f_1 + \cdots + f_i$, we have $\sigma(\boldsymbol{x}) = w$. A count-valued operation can then access the encoding of $w$ using the terms

$$X_i = \overleftarrow{\#}[\overleftarrow{\#}[\boxplus_i] = 0] \qquad \text{for } i = 1, \ldots, n \tag{12}$$

Thus, $X_i$ is the number of positions that have no occurrence of $\boxplus_i$ to their left (and do not carry $\boxplus_i$ themselves). Since there is exactly one occurrence of $\boxplus_i$, this means $X_i$ is exactly the position of $\boxplus_i$, minus one. Therefore, the term $X_i$ evaluates to $\boldsymbol{x}(i)$, meaning we have $\sigma(X_1, \ldots, X_n) = w$.

**Phase II: Simulating the counter machine** During the first phase, our CoT C-RASP appended letters to make an encoding $\boldsymbol{x} \in \mathbb{N}^n$ of the input word available through C-RASP terms Eq. (12). We now use a CM that starts with this encoding in its counters and then decides whether $w \in L$. Such a counter machine exists because of Lemma 3.5 and the fact that $S = \{\boldsymbol{x} \in \mathbb{N}^n \mid \sigma(\boldsymbol{x}) \in L\}$ is recursively enumerable (since $\sigma$ is computable). The simulation of the CM on $\boldsymbol{x}$ works exactly like in Section 3, except that in the terms defined in equation 3, instead of using $\overleftarrow{\#}[Q_{a_i}]$ for $i = 1, \ldots, n$, we use the C-RASP term $X_i$ defined in equation 12. See Appendix C for details.

**Example 2.** *Let us illustrate the case of the language $L = \{a, b\}^*b$ of words that end in $b$. We will need a CM that recognizes the set $S = \{\boldsymbol{x} \in \mathbb{N}^2 \mid \sigma(\boldsymbol{x}) \in L\}$ of encodings of words in $L$. Observe that $\boldsymbol{x} \in \mathbb{N}^2$ satisfies $\sigma(\boldsymbol{x}) \in L$ if and only if $\boldsymbol{x}(1)$ is even: This is because for $x \in \mathbb{N}$ where $\beta(x)$ is non-empty, the string $\beta(x) \in \{0, 1\}^*$ ends in $0$ if and only if $x$ is even. Therefore, our CM in Fig. 1 recognizes exactly $S$. Thus, our CoT C-RASP will have the following rules. For Phase I, it has the rules (7) to (12), where $a_1 = a$ and $a_2 = b$. For Phase II, we want to simulate the CM from Example 1, and so we introduce the same rules as (5) and (6), except that in (6), $Q_a$ is replaced with $X_1$ everywhere. This way, we simulate the CM in Fig. 1 on some encoding $\boldsymbol{x} \in \mathbb{N}^2$ of the input $w$ (i.e. $\sigma(\boldsymbol{x}) = w$) and then check whether $\boldsymbol{x}(1)$ is even.*

## 5 EMPIRICAL EXPERIMENTS

We empirically validate our Turing-completeness results on several complex arithmetic concepts. See Table 2 in Appendix D for details. Our theory predicts that CoT C-RASP with NoPE suffices for unary representation (of numbers), while RPEs are needed for binary representation. Accordingly, we conduct three experiments: 1) *Unary* without positional encodings, 2) *Binary* with RPEs, and 3) *Binary* without RPEs. For each task, we construct two counter machines (CMs), one for the *Unary* representation and one for the *Binary* representation.

We employ a decoder-only LLaMA architecture Touvron et al. (2023) [2] and train all weights from scratch without any pre-trained initialization. The model is trained on inputs of length [1-100] and evaluated on three test sets: an in-distribution split with lengths [1-100] ($test_0$), and two out-of-distribution splits with lengths [101-200] ($test_1$) and [201-300] ($test_2$). The SMATs are trained using AdamW (weight decay 0.01) with a batch size of 64 and maximum 30k steps. To prevent overfitting, we use an EarlyStopping callback that monitors validation loss and stops training if the model's accuracy reaches 100% on the in-distribution test set ($test_0$) for three consecutive epochs.

As shown in Table 1, SMAT achieves strong in-distribution performance on *Unary* representations, with accuracy exceeding 99.90%. It also generalizes well to longer sequences, maintaining high accuracy. In contrast, the *Binary* representation with RPEs exhibits near-perfect generalization across all three test splits, consistently achieving 100% accuracy. However, removing RPEs causes generalization to break down: only Prime reaches around 95% on $test_0$, and all tasks exhibit almost no generalization. Together, these results show a clear contrast: *Unary* inputs generalize naturally with NoPE, whereas *Binary* inputs require RPEs to achieve any meaningful length generalization.

| Language | Unary | | | Binary (w/ RPE) | | | Binary (w/o RPE) | | |
|---|---|---|---|---|---|---|---|---|---|
| | $test_0$ | $test_1$ | $test_2$ | $test_0$ | $test_1$ | $test_2$ | $test_0$ | $test_1$ | $test_2$ |
| Prime | 100 | 100 | 100 | 100 | 100 | 100 | 95.00 | 0.40 | 0.00 |
| Exponential | 99.95 | 99.96 | 99.96 | 100 | 100 | 100 | 82.80 | 0.06 | 0.00 |
| Division | 99.90 | 100 | 99.99 | 100 | 100 | 100 | 76.40 | 0.02 | 0.00 |
| Greatest Common Divisor | 99.99 | 100 | 99.70 | 100 | 100 | 100 | 70.20 | 0.03 | 0.00 |
| Multiplication | 99.99 | 100 | 99.98 | 100 | 100 | 100 | 64.40 | 0.02 | 0.00 |

Table 1: Generalization accuracy on three test sets ($test_0, test_1, test_2$) in unary/binary.

## 6 CONCLUDING REMARKS

**Related work.** Our work builds on Huang et al. (2025): They defined a learnable framework of softmax attention transformers (called Limit Transformers), and a declarative framework (C-RASP) for them. Most of our main results use new techniques that have not been used in relation to transformers, e.g., simulation of counter machines. In relation to the learnability framework, Huang et al. (2025) dealt with transformers without CoT or Relative Positional Encodings, not sufficient for Turing-completeness. We extended their proof techniques to these extensions. Yang et al. (2026) showed that the learnability framework from Huang et al. (2025) is learnable in the limit, but not in a computable sense of Chen et al. (2025). Sälzer et al. (2026) proved that softmax Transformers are "almost" Turing-complete: their projections yield all recursively enumerable languages.

Similar to our work, Hou et al. (2025) aims to provide length-generalizing constructions for Turing completeness. However, there are two key differences. First, we demonstrate the existence of softmax transformer constructions, whereas Hou et al. (2025) only demonstrated constructions in RASP (Weiss et al., 2021). Second, the approach of Hou et al. (2025) ensures length generalization only if no $n$-grams are repeated, for some fixed $n$, which is likely to be unrealistic in the limit of long inputs. In contrast, our approach theoretically ensures full length generalizability.

**Future work.** Recent results have refined Turing-completeness for transformers (albeit with hard attention) by relating the number of CoT steps and complexity classes, e.g., see (Merrill & Sabharwal, 2024) and (Li & Wang, 2025). Refining our result with complexity is left for future work.

---

[2] https://huggingface.co/meta-llama

ACKNOWLEDGMENTS

We thank Pablo Barcelo, Michael Benedikt, David Chiang, Andy Yang, Will Merrill, and Jon Rawski for helpful discussions. This material is based in part upon work supported by the European Union[3] ▪ (ERC, LASD, 101089343, https://doi.org/10.3030/101089343 and and ERC, FINABIS, 101077902, https://doi.org/10.3030/101077902).

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

## A   ADDITIONAL MATERIAL ON SECTION 2

### A.1   FORMAL DEFINITION OF SOFTMAX TRANSFORMERS.

Our definition of softmax transformers follows that of Huang et al. (2025), though we use a highly simplified notation here for exposition. In a SoftMax Averaging Transformers (SMAT), given a sequence

$$\mathbf{v}_1, \ldots, \mathbf{v}_n$$

a single layer outputs

$$\mathbf{w}_1, \ldots, \mathbf{w}_n$$

where

$$\mathbf{w}_i := \mathbf{v}_i + C(\mathbf{v}'_i)$$

where $C(\cdot)$ is a feedforward network, $\mathbf{v}'_i := \sum_{j=1}^{i} \bar{w}(j)\mathbf{v}_j$ and

$$\bar{w} = \text{softmax}(\log n \cdot \{\mathbf{v}_j^T \mathbf{K}^T \mathbf{Q} \mathbf{v}_i\}_{j=1}^{i}) \tag{13}$$

where $\mathbf{v}_i$ denotes activations at position $i$, and $\mathbf{K}$, $\mathbf{Q}$ transform these to keys and queries, respectively. Here, scaling with $\log n$ is included, as it is needed to theoretically represent sparse functions across unboundedly input strings and circumvent theoretical limitations of soft attention (Chiang & Cholak, 2022; Edelman et al., 2022). Here, we show the case of a single head, extension to multiple heads is straightforward.

We assume $C$ is a one-layer feedforward layer, where each hidden unit has either ReLU or Heaviside activation. Here, as in Huang et al. (2025), Heaviside is needed to theoretically represent functions with sharp thresholds; at any finite input length, it can be arbitrarily closely approximated using ReLU MLPs.

Huang et al. (2025) also assume that attention logits are rounded to fixed precision; we do not require this for our results here. Also, whereas Huang et al. (2025) consider Absolute Positional Encodings (APE), which necessitated introducing fixed context windows and positional offsets, we do not consider APE here, and so do not need to introduce offsets. Thus, SMATs considered in the present paper are uniformly applicable to arbitrarily long inputs.

To interface SMAT with an input string $w \in \Sigma^+$, we apply a token embedding function $\text{em} : \Sigma \to \mathbb{R}^k$ for some dimension $k$; these are followed by some number of SMAT layers. To define a CoT SMAT, we need the transformer to be able to output a token. To this end, we define an output function $o : \mathbb{R}^d \to \Sigma$, parameterized by applying a linear function $\mathbb{R}^d \to \mathbb{R}^{|\Sigma|}$ followed by an argmax selecting the symbol receiving the highest score.

Overall, we view an SMAT as a length-preserving map $T : \Sigma^* \to \Sigma^*$, where $T(x)_i$ indicates the symbol predicted after reading the prefix $x_1 \ldots x_i$.

**Discussion** Our formalization of SMAT follows the setting of Huang et al. (2025), which was designed to study the learnability of transformers. We note two aspects, which are needed to enable softmax transformers to represent functions across arbitrarily long inputs, and overcome well-known theoretical limitations of softmax attention (Hahn, 2020; Chiang & Cholak, 2022). First, scaling attention logits with $\log n$ is necessary to represent sparse attention to specific positions, which otherwise would be impossible to achieve using softmax attention (Hahn, 2020; Chiang & Cholak, 2022; Edelman et al., 2022). Importantly, this scaling does not involve any new learnable parameters. Second, using Heaviside activations is necessary to represent functions with sharp thresholds, as is needed to perform exact comparison of counts across unboundedly long lengths. At any finite input length, Heaviside can be arbitrarily closely approximated using ReLU MLPs. We view Heaviside (which is not differentiable) as a theoretical proxy for steep ReLU network as is standardly trainable.

## A.2 PROOFS FOR COT EXPRESSIVENESS AND LEARNABILITY

*Proof of Proposition 2.1.* This is a simple extension of Theorem 9 in Huang et al. (2025), as we now explain.

We define a CoT as a map $\Sigma^* \to \Sigma^*$ from an input string $w \in \Sigma^*$ to the sequence $w_2 \dots, w_N$ generated by a CoT C-RASP or CoT SMAT on the input string $w$. Starting from a CoT generated by a CoT C-RASP program, we aim to translate it to a CoT generated by a CoT SMAT.

We first explain the case without RPEs. We need to show that, if a CoT is generated in C-RASP CoT, then there is an SMAT generating the same CoT. In the case of language acceptance by a single binary label computed at the final token, Theorem 9 in Huang et al. (2025) shows that C-RASP can be simulated by a *limit transformer* without positional information. Our first observation is that, in the model of Huang et al. (2025), a limit transformer without positional information is equivalent to a standard transformer without positional encodings and infinite context window, which in turn is equivalent to an SMAT as defined in our paper here. The proof of Theorem 9 in Huang et al. (2025) builds a transformer that computes the values of all boolean predicates computed in the C-RASP program at each position in the string, with one dimension in the model's activations fo each boolean predicate. This means that the truth values of the expressions $\varphi_{a_i}$ appearing in the switch condition $S$ can also be computed. In order to evaluate the switch condition, we add another layer (whose attention heads have zero value matrices, i.e., don't contribute), then linearly project the relevant entries onto a binary vector of length $|\Gamma|$, and apply a piecewise linear function to convert this into a one-hot vector selecting the lowest-index token $a_i$ such that $\varphi_{a_i}$ is true. We now have a limit transformer which at each position outputs a one-hot vector indicating which CoT token to output. This means, whenever a CoT is expressible in C-RASP CoT, it is also expressible by SMAT with CoT.

We now consider the case with RPEs. We again build on Theorem 9 in Huang et al. (2025). We first note that the definition of attention logits with RPE exactly matches the definition of attention logits in Limit Transformers with functions $\phi$ in Huang et al. (2025), where $\phi(i,j)$ is simply $[\![R]\!](i,j)$. Hence, for the purpose of expressivity, any SMAT[RPEs] transformer is equivalent to a limit transformer. Then, when translating from C-RASP to SMAT, implementing an RPE into an attention head proceeds along exactly the same lines as the translation of the special case $\#[j \leqslant i : \psi(i,j)]P(j)$ in the proof of that theorem. $\square$

*Proof of 2.3.* We first consider the case without RPEs. We build on Theorem 7 in Huang et al. (2025) and its variant for transformers without positional encodings, Corollary 18 in Huang et al. (2025). First, from Proposition 2.1, we know that if a language is expressible in C-RASP CoT, then it is also expressible by SMAT with CoT. The proof of that proposition further notes that our model of SMAT is equivalent to a limit transformer without positional information. Then, by Corollary 18 in Huang et al. (2025), any input-output map expressible by a limit transformer without positional information is length-generalizably learnable. This proves the result for the case without RPEs.

We now consider the case with RPEs. The proof is similar to the previous case; however, we need to (i) show that C-RASP[RPEs] can be simulated by SMATs with RPE, (ii) length generalization for SMAT RPE transformers follows from expressibility by SMATs with RPE. First, regarding (i), we again build on Theorem 9 in Huang et al. (2025), extending our argument from the proof of Proposition 2.1. We first note that the definition of attention logits with RPE exactly matches the

definition of attention logits in Limit Transformers with functions $\phi$ in Huang et al. (2025), where $\phi(i, j)$ is simply $[\![R]\!](i, j)$. Hence, for the purpose of expressivity, any SMAT[RPEs] transformer is equivalent to a limit transformer. Then, when translating from C-RASP to SMAT, implementing an RPE into an attention head proceeds along exactly the same lines as the translation of the special case $\#[j \leqslant i : \psi(i,j)]P(j)$ in the proof of that theorem. Second, regarding (ii), we use Corollary 18 in Huang et al. (2025) and note that the addition of fixed (not learned) RPE to attention heads in both the learned transformers and limit transformers has no impact on the argument. □

## A.3 More on Relative Positional Encodings

Here, we discuss how our formalization of Relative Positional Encodings (RPEs) relates to prior work on RPEs. Recall that we define Relative Positional Encodings (RPEs) as subsets $\mathfrak{R} \subseteq \mathbb{N} \times \mathbb{N}$, defining attention weights as:

$$\bar{w} = \mathrm{softmax}(\log n \cdot \{\mathbf{v}_j^T \mathbf{K}^T \mathbf{Q} \mathbf{v}_i + \underbrace{\lambda [\![\mathfrak{R}]\!](i, j)}_{\text{RPE term}}\}_{j=1}^i). \qquad (14)$$

The key is the RPE term, which adds a position-dependent bias to the attention logits. Here, we interpret $\lambda$ as a bias term and $[\![\mathfrak{R}]\!](i, j)$ as 1 if $(i, j) \in [\![\mathfrak{R}]\!]$; otherwise, it is 0.

Oue formalization abstracts *additive relative positional encodings* (additive RPEs), which add a position-dependent term to the attention logits (Shaw et al., 2018; Dai et al., 2019; Xue et al., 2021; Press et al., 2022; He et al., 2021). Schemes in the literature differ in whether they are parameter-free (e.g., Press et al. (2022)) or involve learnable parameters. We consider the especially simple case where $R$ is determined a-priori, parameter-free, and independent of the task at hand. Here, we review relevant prior work on additive RPEs; we write $q_i := \mathbf{Q}\mathbf{v}_i$ and $k_j := \mathbf{K}\mathbf{v}_j$ for brevity.

1. (Shaw et al., 2018): Here, the RPE term is $q_i^T a_{i-j}$, where $a_{i-j}$ is a learned embedding depending on the relative distance $i - j$ (their Eq. 5).

2. (Dai et al., 2019): Here, the RPE term is $q_i^T r_{i-j} + u^T k_j + v^T r_{i-j}$, where $r_{i-j}$ is a learned embedding depending on the relative distance $i - j$, and $u, v$ are learned global vectors.

3. (Xue et al., 2021): Here, the RPE term is $b_{i-j}$, where $b_{i-j}$ is a learned scalar bias depending on the relative distance $i - j$.

4. (Press et al., 2022): Here, the RPE term is $m \cdot (i - j)$, where $m$ is a learned scalar slope.

5. (He et al., 2021): Here, the RPE term is $q_i^T r_{i-j} + u^T k_j + v^T r_{i-j}$, where $r_{i-j}$ is a learned embedding depending on the relative distance $i - j$, and $u, v$ are learned global vectors. This is very similar to Dai et al. (2019).

Another popular class of RPEs are *multiplicative* RPEs, which transform the key and query vectors with position-dependent matrices (Su et al., 2024). Our RPEs are closest to those of (Xue et al., 2021) and (Press et al., 2022), as they involve adding a scalar bias to the attention logits. Whereas (Xue et al., 2021) learn a separate bias for each possible relative distance, we only require a single $R$ determined a-priori, with no learnable parameters beyond the scalar $\lambda$. In our theoretical analysis, this parameter-free nature is useful for length generalization, ensuring that the number of learned parameters need not increase with the input length.

## A.4 Primer on Huang et al. (2025)

As our results build on Huang et al. (2025), we provide a brief primer on their key definitions and results here. We define both syntax and semantics of C-RASP in the main paper. Here, we provide a simple example, illustrating the formal language $L = \Sigma^* ab \Sigma^*$, taken from Huang et al. (2025):

> **$C - RASP$ program for $L = \Sigma^* ab\Sigma^*$ over $\Sigma = \{a, b\}$ (from Huang et al. (2025))**
>
> | | | |
> |---|---|---|
> | $C_{a-}(i) := \#\left[j \leqslant i, j = i - 1\right] \ Q_a(j)$ | # of immediately preceding $a$ | (1) |
> | $P_{a-}(i) := C_{a-}(i) \geqslant 1$ | Position $i - 1$ holds an $a$ | (2) |
> | $Q_{ab}(i) := Q_b(i) \wedge P_{a-}(i)$ | A substring $ab$ ends at position $i$ | (3) |
> | $C_{ab}(i) := \#\left[j \leqslant i\right] \ Q_{ab}(j)$ | # of substrings $ab$ | (4) |
> | $L(i) := C_{ab}(i) \geqslant 1$ | At least one $ab$ precedes position $i$ | (5) |

We now introduce the key definitions and results from Huang et al. (2025) that we build on. As we focus on No Positional Encodings (NoPE) and Relative Positional Encodings (RPE) transformers, we only define the relevant hypothesis classes here; this makes the analysis easier than in Huang et al. (2025), who also consider APE transformers, which caused a substantial amount of further complexity. In particular, the assumption of "translation invariance" used by Huang et al. (2025) is not needed here.

The idealized learning procedure of Huang et al. (2025) is centered around minimizing a regularizer $\mathcal{R}$ mapping transformers $T$ to numbers, favoring simpler and smaller transformers. It is defined in terms of (i) the number of heads, (ii) the precision used in the transformer's attention computations, (iii) the ranks and norms of the various parameter matrices and vectors. The learning model applies to the class $\mathcal{F}$ of length-preserving functions $f$ mapping strings to sequences of vectors. The idealized learning procedure ("Inference Procedure") is then defined as follows:

**Definition A.1** (Inference Procedure, from Huang et al. (2025)). *Given a function $f \in \mathcal{F}$, the* Inference Procedure *obtains a sequence of transformers $T_1, T_2, \ldots$ as follows. Define $U_n$ as the set of transformers matching the behavior of $f$ on all inputs of length $\leqslant \frac{n}{2}$. Then choose $T_n \in U_n$ such that*

$$\mathcal{R}(T_n) \leqslant \frac{1}{n} + \inf_{T \in U_n} \mathcal{R}(T) \tag{6}$$

Here, the term $\frac{1}{n}$ is used because the class $U_n$ is infinite and the infimum may not be attained; approximate minimization of the regularizer is sufficient. Depending on whether we consider NoPE or RPE transformers, the transformers $T_n$ are taken from the corresponding hypothesis class with NoPE or RPE.

Huang et al. (2025) then show that length generalization in this learning model is equivalent to expressibility by a class of idealized transformers called Limit Transformers. As we focus on the NoPE and RPE cases, the result simplifies to the following statement:

**Theorem A.2** (Guaranteed Length Generalization in the Limit, simplified from Huang et al. (2025)). *Let $f \in \mathcal{F}$. Then the following are equivalent:*

1. *$f$ is expressible by a single transformer that computes $f$ across all input lengths (NoPE or RPE).*

2. *(Guaranteed Length Generalization) Applying the Inference Procedure from Definition A.1 (either in the NoPE or RPE setup, matching the encoding in (1)) to $f$ generates a sequence $T_1, T_2, \ldots$ with $\sup_{n=1,2,3,\ldots} \mathcal{R}(T_n) < \infty$, for which there is some $N_0$ such that, for all $m > N_0$, $T_m$ matches $f$ on all inputs of any length $k \leqslant m$.*

These definitions and results concern an idealized learning procedure that assumes that all data up to input length $\frac{n}{2}$ is fitted perfectly for training; recent follow-up work has expanded by providing more quantitative analyses when only finite data is available (Chen et al., 2025; Izzo et al., 2025). Huang et al. (2025) further provide a translation from C-RASP to transformers, which we build on in our results.

## B  ADDITIONAL MATERIAL ON SECTION 3

In this subsection, we prove Proposition 3.3 from Proposition 3.4.

Suppose $\Sigma = \{a_1, \ldots, a_n\}$. If $L \subseteq a_1^* \cdots a_n^*$ is recursively enumerable, then so is the language $K = \{u \in \Sigma^* \mid \exists v \in L \colon \Psi(u) = \Psi(v)\}$ of all permutations of $L$. Moreover, $K$ is permutation-invariant,

and thus recognized by a CoT C-RASP according to Proposition 3.4. Since $L = K \cap a_1^* \cdots a_n^*$, to turn that CoT C-RASP into a CoT C-RASP for $L$, it remains to check that the input word belongs to the set $a_1^* \cdots a_n^*$. Therefore, for all rules $O_a \leftarrow P$, where $P$ is a C-RASP expression, we use

$$O_a \leftarrow P \wedge \bigwedge_{1 \leqslant i < j \leqslant n} \overleftarrow{\#}[Q_{a_i} \wedge \overleftarrow{\#}[Q_{a_j}] > 0] = 0,$$

where the second conjunct says that there are no positions carrying an $a_i$ that have at least one $a_j$ with $j > i$ to their left. Then, the modified C-RASP clearly recognizes $K \cap a_1^* \cdots a_n^* = L$.

## C  ADDITIONAL MATERIAL ON SECTION 4

**Details of Phase II**   In this section, we present the details of Phase II of the construction in Section 4. For this, first observe that

$$S = \{\boldsymbol{x} \in \mathbb{N}^n \mid \sigma(\boldsymbol{x}) \in L\}$$

is recursively enumerable, since the partial function $\sigma$ is computable. Therefore, by Lemma 3.5, there is a $(n+3)$-counter machine $(P, \Delta, q_0, F)$ such that for any $\boldsymbol{x} \in \mathbb{N}^n$, we have $\boldsymbol{x} \in S$ if and only if from the configuration $(q_0, \boldsymbol{x}, 0, 0, 0)$, the counter machine eventually reaches a control state in $F$.

We simulate a step of the counter machine using the following rule. If the CoT C-RASP finds the letter $\tau$ as the last letter, then for each possible next transition $\tau'$, it checks whether its guard $\varphi_{\tau'}$ is satisfied, and if so, executes $\tau'$ by outputting $\tau'$. Thus, we have

$$O_{\tau'} \leftarrow \varphi_{\tau'}(t_1, \ldots, t_{n+3}) \wedge Q_\tau$$

for any two transitions $\tau, \tau' \in \Delta$ for which $\mathrm{tgt}(\tau) = \mathrm{src}(\tau')$. Here, $t_1, \ldots, t_{n+3}$ are the following terms:

$$t_i = X_i + \sum_{\rho \in \Delta} \boldsymbol{u}_\rho(i) \cdot \overleftarrow{\#}[Q_\rho] \qquad \text{for } i = 1, \ldots, n, \text{ and}$$

$$t_i = \sum_{\rho \in \Delta} \boldsymbol{u}_\rho(i) \cdot \overleftarrow{\#}[Q_\rho] \qquad \text{for } i = n+1, n+2, n+3,$$

where $X_i$ is the count-valued C-RASP term from (12). For $i \in \{n+1, n+2, n+3\}$, $t_i$ is just the sum of counter effects on counter $i$. Equivalently, $t_i$ is the current value of counter $i$ after executing all these transitions. For $i \in [1, n]$, $t_i$ we also add $X_i$, which has the effect that the counters $1, \ldots, n$ are initialized with $X_i$.

Finally, our CoT C-RASP accepts if the output symbol is any $\tau \in \Delta$ with $\mathrm{tgt}(\tau) \in F$.

**Other Proofs**

*Proof of Lemma 4.2.* If $L$ is recognized by a CoT C-RASP, then it is also recognized by an SMAT C-RASP by Lemma 2.1. In fact, our model of SMAT is equivalent to the NoPE special case of the Limit Transformers of Huang et al. (2025). Now Theorem 12 in Huang et al. (2025) shows the following: Take any $k$. For each string $w \in \Sigma^*$, let $F(w) \in \Gamma^* \cup \Gamma^\omega$ be the associated CoT by which the language is recognized via an SMAT. Assume Alice has access to the prefix of $wF(w)$ of length $k$, and Bob has access to the remainder, then Alice needs to communicate just $\mathcal{O}(\log k)$ bits to allow Bob to compute the output of the SMAT at all positions $k+1, k+2, \ldots$. In fact, Theorem 12 in Huang et al. (2025) is stated for the special case where $k$ is half the input length, but the argument is entirely general, as it only relies on the length of Alice's part.

Note that, if the CoT terminates before $k - |w|$ steps, Alice can just communicate that. Now given the SMAT recognizes $L$ via CoT, Bob can determine[4] from Alice's communication if a given string is in the language or not.

Now we construct a family of NFAs accepting the language as follows.

---

[4]This is not decidable, but Bob in this model is a computationally unconstrained agent, with communication between Alice and Bob as the only bottleneck.

For $x, y \in \Sigma^*$, define $x \equiv_{AB} y$ if and only if, for all $z \in \Sigma^*$, Alice communicates the same to Bob on $xz$ and $yz$. By definition, each equivalence class of this relation is a subclass of a Nerode equivalence class of $L$ (†).

Given any length bound $n \in \mathbb{N}$, let $Q_n$ be the set of all $\equiv_{AB}$-classes represented by at least some words of length $\leqslant n$. By the result described above, $|Q_n|$ is bounded by $\leqslant \sum_{k=1}^{n} 2^{\mathcal{O}(\log k)} = \mathcal{O}(poly(n))$. Now, by definition of the congruence, $Q_n$ is the state set of an automaton computing $\equiv_{AB}$-equivalence classes. By (†), it recognizes $L$.

We remark that this proof also applies even if we augment transformers with learned Absolute Positional Encodings (APE), because Theorem 12 in Huang et al. (2025) also applies to the APE-augmented case when the APE embeddings are learned. □

# D ADDITIONAL MATERIAL ON SECTION 5

## D.1 DATASET CONSTRUCTION

For each task shown in Table 2, we generate paired datasets of input strings and $k$-CM output traces under two encoding regimes: *Unary* and *Binary* encoding.

| Language | Unary Representation | Binary Representation |
|---|---|---|
| Prime | $\{ a^p : p \in \mathbb{P} \}$ | $\{ \mathrm{bin}(p) : p \in \mathbb{P} \}$ |
| Exponential | $\{ a^{2^i} : i \geqslant 0 \}$ | $\{ \mathrm{bin}(i)\#\mathrm{bin}(j) : j = 2^i \}$ |
| Division | $\{ a^i b^j : j \mid i \}$ | $\{ \mathrm{bin}(i)\#\mathrm{bin}(j) : j \mid i \}$ |
| Greatest Common Divisor | $\{ a^i b^j c^k : k = \gcd(i,j) \}$ | $\{ \mathrm{bin}(i)\#\mathrm{bin}(j)\#\mathrm{bin}(k) : k = \gcd(i,j) \}$ |
| Multiplication | $\{ a^i b^j c^k : k = i \cdot j \}$ | $\{ \mathrm{bin}(i)\#\mathrm{bin}(j)\#\mathrm{bin}(k) : k = i \times j \}$ |

Table 2: *Unary* and *Binary* representation of arithmetic languages. Here $\mathbb{P}$ is the set of prime numbers, $j \mid i$ denotes divisibility, $\gcd(i,j)$ is the greatest common divisor, and $i \times j$ is multiplication.

**Unary Encoding.** In the unary setting, we work over small alphabets such as $\{a\}$ for Prime, $\{a, b\}$ for Exponential, and Division and $\{a, b, c\}$ for Greatest Common Divisor and Multiplication. Here, input strings $w$ are sampled uniformly at random from these alphabets within given length ranges, without enforcing that they encode tuples of integers satisfying the intended arithmetic relation (e.g. words are not constrained to be of the form $a^i b^j c^k$).

Given a deterministic $k$-counter machine (or $k$-CM)

$$M = (P, \Delta, q_0, F),$$

and a unary word $w \in \Sigma^*$, we view $w$ simply as an *input* to $M$. Since $M$ is deterministic, the run of $M$ on $w$ is uniquely defined. Writing $w = w_1 w_2 \cdots w_{|w|}$, the induced computation is the sequence

$$(q_0, \mathbf{c}_0) \xrightarrow{w_1} (q_1, \mathbf{c}_1) \xrightarrow{w_2} \cdots \xrightarrow{w_{|w|}} (q_{|w|}, \mathbf{c}_{|w|}),$$

where $(q_t, \mathbf{c}_t)$ denotes the configuration after reading the $t$-th symbol of $w$.

For a transition $\tau = (p, \varphi, q, u) \in \Delta$, we use the standard notation $\mathrm{src}(\tau) := p$, $\mathrm{tgt}(\tau) := q$, $\varphi_\tau := \varphi$, and $u_\tau := u$. The *target sequence* associated with $w$ is then defined as

$$\mathrm{target}(w) := \left( \tau_t \right)_{t=1}^{|w|},$$

where $\tau_t$ is the unique transition of $M$ taken at step $t$ of the above run. Because $M$ is deterministic, the sequence $\mathrm{target}(w)$ is well-defined and uniquely determined by $w$.

**Binary Encoding.** In the binary setting, integers are represented in canonical binary form (with no leading zeros), over alphabets $\Sigma \in \{\{0, 1\}, \{0, 1, /\}\}$. For the tasks Greatest Common Divisor and Multiplication, we construct inputs of the form $\mathrm{bin}(x) \,/\, \mathrm{bin}(y) \,/\, \mathrm{bin}(z)$, while Exponential and Division use binary pairs $\mathrm{bin}(w) \,/\, \mathrm{bin}(v)$, and Prime uses a single binary encoding $\mathrm{bin}(n)$.

Each binary sample is labelled *positive* when the intended arithmetic relation holds (e.g., $z = x + y$, $z = x \cdot y$, $z = \gcd(x, y)$, $w \mid v$, $z = x^y$, or $n$ is prime). Negative samples are generated by replacing the *input* component with a nearby but incorrect integer that satisfies the required bit-length constraints.

As in the unary setting, the *input string* is fed directly to the model, and the *supervision signal* is given by the $K$-CM trace obtained by running the corresponding deterministic $k$-CM on this binary input; thus the target sequence is uniquely defined.

## D.2 DETAILS OF EXPERIMENTAL SETUP

**Prompt and Predicted Output.** For every input string $w$, we prepare the model input in a prefix–LM format. The model receives the prompt $\boxed{\text{SOS} \mid \text{INPUT} \mid \text{SEP}}$ where INPUT denotes either the unary or binary representation of the original string $w$. After the separator token, the model is required to autoregressively generate the target region $\boxed{\text{TARGET} \mid \text{EOS}}$ where TARGET encodes $\tau(w)$, the unique accumulator trace produced by the deterministic $k$-CM when executed on $w$. Thus the complete input–target sequence used during training has the form $\boxed{\text{SOS} \mid \text{INPUT} \mid \text{SEP} \mid \text{TARGET} \mid \text{EOS}}$.

During training, we apply the standard autoregressive language modeling objective, but we restrict the cross-entropy loss to the TARGET region (TARGET--EOS), ensuring that the model learns to generate the target trace $\tau(w)$ conditioned on the INPUT prefix. At evaluation time, we report exact match (EM) over the entire predicted output region: an example receives score 1 if the model's generated sequence matches $\tau(w)$ exactly, and 0 otherwise.

**Architecture and hyperparamters** All models in this work are trained *from scratch*, without any pretrained weights. We use a decoder-Only Transformer architecture *LLaMA*, but with the standard SwiGLU activation replaced by a ReLU nonlinearity in all feed-forward blocks. Beyond the activation change, we also modify the positional encoding mechanism: the *Unary* representation uses NoPE, whereas the *Binary* representation uses our relative positional encodings. Apart from these substitutions, the model follows the standard LLaMA design, including multi-head self-attention, layer normalization, and residual connections. Our empirical results show that the architecture performs robustly under both the *Unary* and *Binary* encodings considered in this work.

The hyperparameters used for each task are listed in Table 3, including the number of layers, attention heads, embedding dimension, learning rate, and maximum training steps.

| Language | Representation | Model Size | LR | Max Steps |
|---|---|---|---|---|
| | Unary | 1 layer; 1 head; 32 dim | 1e–3 | 30k |
| Prime | Binary$^{\mathfrak{R}}$ | 1 layer; 1 head; 64 dim | 1e–3 | 30k |
| | Binary$^{\mathcal{N}}$ | 6 layer; 4 head; 256 dim | 1e–3 | 30k |
| | Unary | 1 layer; 1 head; 32 dim | 1e–3 | 30k |
| Exponential | Binary$^{\mathfrak{R}}$ | 1 layer; 1 head; 64 dim | 1e–3 | 30k |
| | Binary$^{\mathcal{N}}$ | 6 layer; 4 head; 256 dim | 1e–3 | 30k |
| | Unary | 4 layer; 2 head; 128 dim | 1e–3 | 30k |
| Division | Binary$^{\mathfrak{R}}$ | 1 layer; 1 head; 64 dim | 1e–3 | 30k |
| | Binary$^{\mathcal{N}}$ | 6 layer; 4 head; 256 dim | 1e–3 | 30k |
| | Unary | 3 layer; 1 head; 128 dim | 1e–3 | 30k |
| Greatest Common Divisor | Binary$^{\mathfrak{R}}$ | 1 layer; 1 head; 64 dim | 1e–3 | 30k |
| | Binary$^{\mathcal{N}}$ | 6 layer; 4 head; 256 dim | 1e–3 | 30k |
| | Unary | 3 layer; 1 head; 64 dim | 1e–3 | 30k |
| Multiplication | Binary$^{\mathfrak{R}}$ | 1 layer; 1 head; 64 dim | 1e–3 | 30k |
| | Binary$^{\mathcal{N}}$ | 6 layer; 4 head; 256 dim | 1e–3 | 30k |

Table 3: Hyperparameters used for training LLaMA-style decoder-only Transformers on each task, across the *Unary* (NoPE) and *Binary* (Binary$^{\mathfrak{R}}$ with RPEs, Binary$^{\mathcal{N}}$ without RPEs) representations. All models use ReLU activations and are trained from scratch with AdamW. Weight decay is $0.01$ for Prime, Exponential, and GCD; $0.05$ for Division; and $0.03$ for Multiplication.

