# OpenReview forum: "Softmax Transformers are Turing-Complete"
_ICLR.cc/2026/Conference — ICLR 2026 Oral_

### Official Review · Reviewer_DApH · 2025-10-23

**Soundness:** 3
**Presentation:** 3
**Contribution:** 3
**Rating:** 6
**Confidence:** 3

**Summary:**

The paper proves, for the first time, that soft attention transformers with chain-of-thought (CoT) are Turing-complete. This is more realistic than past results, which assumed hard attention. The authors use C-RASP to demonstrate that soft attention transformers with CoT can simulate Minsky counter machines, which are known to be equivalent to Turing machines. They show that such transformers are Turing-complete for unary and letter-bounded languages, and that adding relative positional encodings (RPEs) extends this result to arbitrary languages. Empirical results on arithmetic reasoning tasks confirm the theoretical claims: transformers with RPEs achieve near-perfect accuracy and have good length generalization.

**Strengths:**

1. The paper resolves a major open question in the theoretical literature of transformer expressivity. This result brings us closer to understanding what CoT transformers can perform in practice.

2. The proofs and constructions are rigorous and technically sound. Moreover, the main result is proven using a new proof technique---using counter machines---which strengthens the theoretical contribution.

3. The paper includes experiments on arithmetic reasoning tasks, with results that align closely with the theory: unary tasks succeed without RPEs, while binary tasks require RPEs to generalize.

**Weaknesses:**

1. Contribution (ii) from the introduction claims to provide a guarantee on trainability; however, this seems somewhat misleading, as the paper does not present a formal analysis of learnability.

2. In the current form, the paper is quite technical and, in places, hard to follow unless the reader is very familiar with related work. I believe the paper would benefit from some more high-level intuitive explanations. Additionally, a summary figure outlining the main results and their corresponding sections would improve clarity.

3. I think the experimental setup would benefit from some more detail and polishing. For instance, there is no mention of the test sets.

**Questions:**

1. How would the results of Merrill and Sabharwal (2024) translate to this setup?

2. Do you think it would be possible to generalize the result to other types of positional encodings?

3. Can you comment on the val_2 accuracy only approaching, but not reaching 100%? What does this imply? Could you say that the models learn a very good approximation but maybe not actually learn the underlying mechanism? Did you perform any ablation studies to see if you can improve the reported results? I think some discussion about this should be included in the paper.


William Merrill and Ashish Sabharwal. 2024. The Expressive Power of Transformers with Chain of Thought.

---

> ### Author Response · Authors · 2025-11-20
>
> Dear reviewer DApH,
>
> Thanks for your feedback on our paper!
>
> ### Reply Regarding Weaknesses
>
> > (1) Contribution (ii) from the introduction claims to provide a guarantee on trainability; however, this seems somewhat misleading, as the paper does not present a formal analysis of learnability
>
> Our results do provide a guarantee of trainability in the sense of length generalizability (in the sense of Huang et al). Roughly speaking, a language $L$ is length-generalizable if an idealized learning procedure for transformers converges to $L$, if provided with all strings from $L$ up to length $i$ (for some $i$) in the training. To obtain such a guarantee, we proved simple extensions of it. to the setting of CoT and Relative Positional Encodings (Proposition 2.1 and Proposition 2.3). We added short proof ideas in the main body (the longer versions are in the appendix, owing to lack of space). In addition, to make the paper more self-contained, we have added a new section in the appendix (Appendix A.4), which discusses the idealized learning procedure of Huang et al in more detail.
>
>
> > (2) In the current form, the paper is quite technical and, in places, hard to follow unless the reader is very familiar with related work. I believe the paper would benefit from some more high-level intuitive explanations. Additionally, a summary figure outlining the main results and their corresponding sections would improve clarity.
>
> We have added a roadmap at the beginning of Section 3 and simple examples for our main CoT transformer constructions (i.e. in Section 3 and Section 4) to clarify how our main constructions work. For Section 2, we have also added some more intuitive explanations of the learning setting/guarantee (more details added into the appendix).
>
>
> > (3) I think the experimental setup would benefit from some more detail and polishing. For instance, there is no mention of the test sets.
>
>  We revised Section 5, where we polished the writing, added more details, and reported the latest results. Owing to space limit, some details of the experiment setup and dataset construction etc, have been relegated into Appendix D.
>
>
> ### Reply Regarding Questions
>
> > (1) How would the results of Merrill and Sabharwal (2024) translate to this setup?
>
>  If we understand you correctly, you are asking for a clarification on the connection between our results and Merrill and Sabharwal (2024). Firstly, they provided a simulation of Turing machines using hard attention transformers (more precisely, AHAT - Averaging Hard Attention Transformers), i.e., not softmax transformers. Learnability is also not considered in their work. Finally, we provide empirical results on training CoT transformers recognizing languages that require non-trivial arithmetic reasoning.
>
> Merrill et al also provided a precise correspondence between time complexity and CoT AHATs, i.e., a single step of a Turing machine can be simulated by a single step of CoT AHATs. We have mentioned this as an open problem in the case of CoT softmax transformers. Our simulation captures k steps of counter machines with k CoT steps. However, counter machines could require exponentially many more steps to simulate Turing machines. It is unclear if CoT softmax transformers could be as "efficient" as CoT AHATs.
>
> > (2) Do you think it would be possible to generalize the result to other types of positional encodings?
>
> Besides RPEs, the main other type of positional encodings is Absolute Positional Encodings (APE). Indeed, SMATs with learnable APE (as used e.g. in GPT-2) will not be Turing-complete in a length-generalizable manner over a binary alphabet. The reason is that Huang et al shows that length generalization for learnable APE is only possible for tasks satisfying the conclusion of our Lemma 4.2. This means that CoT SMAT with learnable APE cannot be Turing complete, for the same reason that CoT SMAT without positional encodings cannot.
>
> > (3) Can you comment on the val_2 accuracy only approaching, but not reaching 100%? What does this imply? Could you say that the models learn a very good approximation, but maybe not actually learn the underlying mechanism? Did you perform any ablation studies to see if you can improve the reported results? I think some discussion about this should be included in the paper.
>
> In our previous experiments, models trained with the default hyperparameter configuration exhibited high but not 100% accuracy. These results confirmed that length generalization succeeded for unary representation with NoPE and binary representation for RPEs. We had conducted ablation studies on positional encodings for the binary representation, comparing models trained with RPEs against those without. In the new version, we systematically varied model capacity to find the best hyperparameters. The updated results are reported in Table 2, and the detailed experimental setup is provided in Appendix D.

---

> > ### Comment · Reviewer_DApH · 2025-11-25
> >
> > I thank the authors for their detailed response and for updating the draft. I believe that, in the current form, the paper is much stronger; the authors did a good job at adding the necessary clarifications to improve clarity and readability.
> >
> > Given that the rebuttal addressed all of my comments and questions, and, as far as I can tell, also the concerns of the other reviewers, I will raise my score to 8.

---

> > > ### Author Response · Authors · 2025-11-28
> > >
> > > Thank you very much for upgrading your score.

---

### Official Review · Reviewer_b98p · 2025-10-31

**Soundness:** 2
**Presentation:** 2
**Contribution:** 2
**Rating:** 2
**Confidence:** 3

**Summary:**

This paper proves the completeness of softmax Transformers for CoT C-RASP over a unary alphabet. The result also implies that softmax Transformers exhibit length generalization for this language. The authors further state that softmax Transformers are not Turing-complete for arbitrary languages but show that softmax Transformers with relative positional encoding are Turing-complete for arbitrary languages.

**Strengths:**

The paper consider an important theoretical question, i.e., whether Turing-completeness holds for more realistic models such as softmax Transformers.

**Weaknesses:**

1. The expression "Turing-completeness for some languages" is conceptually unclear. Turing-completeness has a strict formal definition. The results presented in the paper do not appear to fully establish the claim.

2. Softmax Transformers should, in principle, approximate hard-attention arbitrarily well. Since hard-attention Transformers are known to be Turing-complete, the paper’s results suggesting a gap between softmax Transformers and Turing-completeness raise questions. The paper seems to provide no clear explanation for this discrepancy.

**Questions:**

See Weaknesses.

---

> ### Author Response · Authors · 2025-11-20
>
> Dear reviewer b98p,
>
> Thanks for your feedback on our paper!
>
> ### Reply Regarding Weaknesses
>
> > (1) The expression "Turing-completeness for some languages" is conceptually unclear. Turing-completeness has a strict formal definition. The results presented in the paper do not appear to fully establish the claim.
>
> We do satisfy the strict formal definition, and hence fully establish the claim: The Turing-completeness result referred to in the title and the abstract is Theorem 4.3, which proves that every recursively enumerable language can be accepted by a softmax transformer in the CoT setting. We reformulate Theorem 4.3 to make this clearer.
>
> Perhaps the reviewer mistook Prop. 3.3. or Prop. 3.4 as the main result. These are observations on the way to Theorem 4.3, and they are indeed restricted Turing-completeness results: They show that even without relative positional encodings (RPEs), softmax transformers can recognize all recursively enumerable languages that are (i) letter-bounded (in the case of Prop 3.3) or (ii) permutation-invariant (in the case of Prop. 3.4).
>
> > (2) Softmax Transformers should, in principle, approximate hard-attention arbitrarily well. Since hard-attention Transformers are known to be Turing-complete, the paper’s results suggesting a gap between softmax Transformers and Turing-completeness raise questions. The paper seems to provide no clear explanation for this discrepancy.
>
> Turing-completeness results were shown in ([3], [4], etc.) for Averaging Hard Attention Transformers (AHATs). However, experimental and theoretical evidence suggests that AHAT expressiveness does not imply that a function is expressible by softmax attention, let alone that it is learnable. For instance, AHATs are known to be able to express PARITY (e.g., [2]), which is known to not be learnable by softmax transformers [1, 2].
>
> A second aspect is learnability and length-generalization. A key aspect of our results is that they not only provide softmax-based constructions, but also associated guarantees of length-generalizability. As real-world transformers use softmax attention, softmax transformers come with associated theoretical tools for showing length generalization (particularly, the framework of [1], which we use). In summary, our result entails that transformers can actually be trained for *arbitrary recursively enumerable languages* in a way that ensures length-generazability. No trainability guarantee follows from previous Turing-completeness results.
>
> [1] Huang et al, A formal framework for understanding length generalization in transformer, 2025. https://openreview.net/forum?id=U49N5V51rU
>
> [2] Hahn et al, Why are sensitive functions hard for transformers?, 2024. https://arxiv.org/abs/2402.09963
>
> [3] Pérez et al,  Attention is turing-complete,2021. https://dl.acm.org/doi/10.5555/3546258.3546333
>
> [4] Merrill et al, The expressive power of transformers with chain of thought, 2023. https://arxiv.org/abs/2310.07923

---

> > ### Comment · Reviewer_b98p · 2025-11-28
> >
> > I appreciate the authors’ detailed rebuttal. Their response adequately addresses my concern. I will raise my score to 6.

---

> > > ### Author Response · Authors · 2025-11-28
> > >
> > > Thank you for raising your score to 6. We will be happy to answer further questions.

---

### Official Review · Reviewer_rteb · 2025-11-01

**Soundness:** 2
**Presentation:** 2
**Contribution:** 3
**Rating:** 4
**Confidence:** 2

**Summary:**

This paper studies whether softmax-attention Chain-of-Thought (CoT) transformers are Turing-complete and answers this question positively. To prove this, they build upon previous work on Chain-of-Thought extensions of a declarative language called C-RASP. They first show that CoT C-RASPs with causal masking are not Turing complete over arbitrary languages, and therefore extend them to CoT C-RASPs with Relative Positional Encodings. Lastly, they support their theoretical results by training Softmax Attention Transformers on various arithmetic tasks and find that they learn these tasks very well.

**Strengths:**

The paper is able to answer the relevant question whether softmax-attention Chain-of-Thought (CoT) transformers are Turing-complete positively, thereby extending previous work that only showed Turing-completeness using hardmax-attention. Furthermore, it is insightful to see that CoT C-RASPs with causal masking are not generally Turing complete, but that they do become Turing complete when adding Relative Positional Encodings.

**Weaknesses:**

I think the paper could profit from additional clarity in the writing and in the explanations. I was for example slightly confused by phrases like 'to provide _a kind of_ guarantee of trainability' (in the contributions). Furthermore, I found the section on the 'Empirical Experiments' not very clearly written and would have been interested in seeing slightly more details and explanations on the tasks, the architectures used and the training. The corresponding Appendix D was very short and did not help me much. Lastly, the related work section was very short and I would have in particular been interested to see more discussion on the relation between these results and previous results using hardmax-attention and the relation between this paper and the results from Huang et al. (2025), which is cited very often throughout the paper.

**Questions:**

1. What do you mean by 'to provide _a kind of_ guarantee of trainability' (in the contributions)? Could you explain the usage of 'kind of' better here.
2. Could you discuss the usage of RPEs for providing positional information in Transformers in more detail? How often are these used and what have been previous empirical findings on them?
3. Many of your definitions and proofs seem to build upon results from Huang et al. (2025). Could you discuss the relation to this paper in more detail.
4. Could you describe the tasks, training architecture, and loss function used in the 'Empirical Experiments' in slightly more detail again?
5. Where can I see the results that you are referring to in this sentence:
> These results demonstrate that unary benefits from NoPE, whereas binary requires R for length generalization.

---

> ### Author Response · Authors · 2025-11-20
>
> Dear reviewer rteb,
>
> Thanks for your feedback on our paper!
>
> ### Reply Regarding Questions
>
> > (1) What do you mean by 'to provide a kind of guarantee of trainability' (in the contributions)? Could you explain the usage of 'kind of' better here
>
> By trainability, we meant "length generalizability" in [1]. Roughly speaking, a language $L$ is length-generalizable if an idealized training algorithm converges to $L$, if provided with all inputs of $L$ up to some length $i$. In the new version, we have added a clarification just below this paragraph.
> > (2) Could you discuss the usage of RPEs for providing positional information in Transformers in more detail? How often are these used and what have been previous empirical findings on them?
>
> We have added Appendix A.3 to provide further review of prior work on RPEs. In a nutshell, our formalization covers a substantial set of positional encoding schemes from the literature.
>
> > (3) Many of your definitions and proofs seem to build upon results from Huang et al. (2025). Could you discuss the relation to this paper in more detail.
>
> In short, [1] defined a learnable framework of softmax attention transformers (called Limit Transformers), and a declarative framework (C-RASP) for them. In this paper, we further show that these classes of transformers are Turing-complete. It is only in this sense that our results build on top of [1] since most of our main results use new techniques that have not been used in relation to transformers, e.g., simulation of counter machines.
>
> In relation to the learnability framework itself, the results of [1] dealt with transformers without CoT and Relative Positional Encodings, which are not sufficient for Turing-completeness. We extended the proof techniques in [1]. to these extensions (Proposition 2.1 and Proposition 2.3).
>
> > (4, 5) Could you describe the tasks, training architecture, and loss function used in the 'Empirical Experiments' in slightly more detail again?  Where can I see the results that you are referring to in this sentence: These results demonstrate that unary benefits from NoPE, whereas binary requires R for length generalization.
>
> In the revised version, Appendix D now includes a complete description of the dataset construction and our experimental setup. We have additionally revised all tables to present the latest results.
>
>
> [1] Huang et al, A formal framework for understanding length generalization in transformer, 2025. https://openreview.net/forum?id=U49N5V51rU

---

> > ### Comment · Reviewer_rteb · 2025-11-28
> >
> > I thank the authors for answering my questions and addressing my concerns.
> >
> > The main concern that I do not see directly addressed is that while there is now a more detailed primer on Huang et al. (2025) in Appendix A.4, I am still missing a clear discussion of the relation between Huang et al. (2025) and your paper at some point in the paper (similar to the answer you gave me).
> >
> > I do think that the updates made in response to my review and the other reviews improved the paper. I will therefore increase my score to a 6.

---

> > > ### Author Response · Authors · 2025-11-28
> > >
> > > Thank you for increasing your score to 6.
> > >
> > > We have uploaded a new version that discusses the connection to Huang et al. (2025); see Conclusion Section.

---

### Official Review · Reviewer_PWUm · 2025-11-01

**Soundness:** 4
**Presentation:** 2
**Contribution:** 4
**Rating:** 10
**Confidence:** 2

**Summary:**

Previous work had shown that hardmax attention transformers are Turing-complete. (Hardmax attention usually means that activations are 1 for some token and 0 elsewhere.) This paper proves the results under the more realistic case when the softmax operation is used in the network architecture, rather than the hardmax. The result goes through different techniques than the prior literature on Turing-completeness of transformers.

The proof of Turing-completeness, at a high level, involves two steps:

- Show that a language CoT C-RASP can be implemented by a softmax transformer (this mainly seems to have been done in prior work by Huang et al).
- Show that CoT C-RASP is Turing-complete, under various conditions (if the language is letter-bounded or there are relative positional encodings).

**Strengths:**

- This paper takes a step towards proving the Turing-completeness of transformers under more realistic architecture definitions. Softmax (rather than hardmax) is differentiable, so this proof is the first one for a transformer architecture definition that is trainable via gradient methods.
- The proof of Turing completeness goes through different techniques than previous proofs for hardmax attention. I am not familiar enough with these techniques to know if they are standard in formal language theory, but in any case bringing them to the attention of the community focused on computability in LLMs may be useful.

**Weaknesses:**

- Through Section 2, many proofs of the results, definitions etc. seem to taken nearly directly from Huang et al (2025). This makes it hard  to read without first reading Huang et al. There is significant setup assumed from Huang et al, some of which is never described in the paper itself. Generally, it seems worth defining notation before it is used, even if the notation is standard in formal language theory. Notation was not defined for the definition of C-RASP, CoT C-RASP and in many other places.  If this takes too much space, it would be nice to include an appendix with relevant background so that the paper is more self-contained.
- The paper should do more to highlight to conceptual or technical challenges of moving from hardmax to softmax. What about this alternate proof strategy allowed for circumventing the need/convenience of assuming hard attention?

**Questions:**

N/A

---

> ### Author Response · Authors · 2025-11-20
>
> Dear reviewer PWUm,
>
> Thanks for your feedback on our paper!
>
> ### Reply Regarding Weaknesses
>
> > (1) Through Section 2, many proofs of the results, definitions etc. seem to taken nearly directly from Huang et al (2025). This makes it hard to read without first reading Huang et al. There is significant setup assumed from Huang et al, some of which is never described in the paper itself. Generally, it seems worth defining notation before it is used, even if the notation is standard in formal language theory. Notation was not defined for the definition of C-RASP, CoT C-RASP and in many other places. If this takes too much space, it would be nice to include an appendix with relevant background so that the paper is more self-contained.
>
> We have added some intuition in Section 2 and created Appendix A.4 to make the paper self-contained for readers who want to understand the full learning setup/guarantee from [1].
>
> > (2) The paper should do more to highlight to conceptual or technical challenges of moving from hardmax to softmax. What about this alternate proof strategy allowed for circumventing the need/convenience of assuming hard attention?
>
> The main challenge is to perform a direct simulation of Turing machines (as is always done in previous proofs), which seems rather difficult to do using softmax transformers. In particular, it is unclear how to extract the position of the head of the Turing machine using softmax (instead of hard attention plus either -|<x,y>| attention score as done by Perez et al. or layer norm as done by Merrill&Sabharwal'23). The following are our main innovations to overcome the challenge.
>
> First innovation: simulate Minsky's counter machines using C-RASP with CoT, which can be translated to softmax transformers with CoT. To the best of our knowledge, this is the first Turing-completeness proof of transformers that exploits connections to Minsky's counter machines. The key idea here is that softmax can still perform *counting*, which can be captured using softmax transformers (through C-RASP connection, as per [1]). In particular, this also yields a length-generalizability guarantee as a bonus.
>
> Second innovation: Extending C-RASP by RPEs. The use of C-RASP to simulate counter machines alone allows us to prove our first main result about letter-bounded languages. However, for languages that are not letter-bounded, the counting capabilities of C-RASP alone do not suffice: The input word is not necessarily determined by its letter counts (we actually prove Turing-incompleteness of C-RASP CoT in Prop. 4.1). Therefore, we extend C-RASP with relative positional encodings (RPEs), and show that this extension is still length-generalizable.
>
> Third innovation: Using RPEs to compute a numerical encoding of the input word. In our third innovation, we show that RPEs are sufficient for working with arbitrary input words: They allow us to compute an unambiguous encoding of the input word into a number that can be accessed by the simulated counter machine. This way, we prove full Turing-completeness in the presence of RPEs.
>
> We have added a remark in the introduction.
>
> [1] Huang et al, A formal framework for understanding length generalization in transformer, 2025. https://openreview.net/forum?id=U49N5V51rU

---

### Author Response · Authors · 2025-11-28

Dear AC and Reviewers,

We thank you for your valuable feedback. We would like to briefly summarize the current state of the rebuttal. We initially received the scores *10, 6, 4, 2*. After the rebuttal, the scores are currently *10, 8, 6, 6*. We understand the last two scores are still not updated, owing to unexpected circumstances with OpenReview. At any rate, we believe that the current scores (*10, 8, 6, 6*) reflect that we have satisfactorily addressed the reviewers comments.

Once again, we thank you for all the feedback that you have provided, which has certainly strengthened the paper. We will be happy to answer any additional questions.

Best wishes,

The Authors

---

### Meta-Review · Area_Chair_tb5S · 2026-01-08

**Summary:**

This paper investigates the theoretical expressivity of softmax-attention Chain-of-Thought (CoT) transformers, specifically addressing whether they are Turing-complete. While previous work established Turing-completeness for hardmax-attention transformers, the status of the more realistic softmax architecture remained an open question. The authors prove that length-generalizable softmax CoT transformers are indeed Turing-complete. Their approach involves simulating Minsky counter machines via an extension of the Counting RASP (C-RASP) language.

**Reviewer Concerns:**

The reviewers initially raised several significant concerns which the authors addressed during the rebuttal phase:
(1)	Reviewers PWUm and rteb noted a heavy reliance on the framework of Huang et al. (2025), making the paper difficult to read independently. The authors addressed this by adding intuition to Section 2 and creating Appendix A.4 to provide necessary background.
(2)	Reviewer PWUm asked for a better highlight of the technical challenges in moving from hardmax to softmax. The authors clarified their three main innovations: the simulation of Minsky counter machines, the extension of C-RASP with RPEs, and the use of RPEs for numerical encoding of input words.
(3)	Reviewers rteb and DApH requested more details on the training setup, datasets, and test sets. The authors updated Appendix D with complete descriptions of dataset construction and experimental hyperparameters.
(4)	Reviewer b98p expressed concern over the formal definition of "Turing-completeness for some languages". The authors clarified that Theorem 4.3 establishes full Turing-completeness for recursively enumerable languages, while earlier propositions were intermediate steps.

**Reviewer Scores:**

The scores saw a significant upward trend following the rebuttal. Although technical issues with OpenReview initially prevented some reviewers from updating their official score buttons, their intent was clearly stated in the discussion.

Reviewer PWUm: Maintained 10 (Strong Accept).

Reviewer DApH: Raised from 6 to 8 (Accept) after the authors clarified the trainability guarantees and updated the experimental results.

Reviewer rteb: Raised from 4 to 6 (Marginal Accept) after the addition of the Huang et al. primer and the discussion of RPEs.

Reviewer b98p: Raised from 2 to 6 (Marginal Accept) after the authors clarified the discrepancy between hard-attention and softmax-attention regarding learnability and length generalization.

---

### Decision · Program_Chairs · 2026-01-26

Accept (Oral)